# Emergence of novel cephalopod gene regulation and expression through large-scale genome reorganization

Hannah Schmidbaur [1,9], Akane Kawaguchi[2,9], Tereza Clarence[3], Xiao Fu [3], Oi Pui Hoang[1], Bob Zimmermann [1], Elena A. Ritschard[1,4], Anton Weissenbacher[5], Jamie S. Foster [6], Spencer V. Nyholm[7], Paul A. Bates [3], Caroline B. Albertin [8✉], Elly Tanaka [2✉] & Oleg Simakov [1✉]

Coleoid cephalopods (squid, cuttlefish, octopus) have the largest nervous system among invertebrates that together with many lineage-specific morphological traits enables complex behaviors. The genomic basis underlying these innovations remains unknown. Using comparative and functional genomics in the model squid *Euprymna scolopes*, we reveal the unique genomic, topological, and regulatory organization of cephalopod genomes. We show that coleoid cephalopod genomes have been extensively restructured compared to other animals, leading to the emergence of hundreds of tightly linked and evolutionary unique gene clusters (microsyntenies). Such novel microsyntenies correspond to topological compartments with a distinct regulatory structure and contribute to complex expression patterns. In particular, we identify a set of microsyntenies associated with cephalopod innovations (MACIs) broadly enriched in cephalopod nervous system expression. We posit that the emergence of MACIs was instrumental to cephalopod nervous system evolution and propose that microsyntenic profiling will be central to understanding cephalopod innovations.

[1] Department of Neurosciences and Developmental Biology, University of Vienna, Vienna, Austria. [2] Institute for Molecular Pathology, Vienna, Austria. [3] Biomolecular Modelling Laboratory, The Francis Crick Institute, London, UK. [4] Department of Biology and Evolution of Marine Organisms, Stazione Zoologica Anton Dohrn, Naples, Italy. [5] Vienna Zoo, Maxingstraße 13b, 1130 Vienna, Austria. [6] Department of Microbiology and Cell Science, University of Florida, Space Life Science Lab, Merritt Island, FL, USA. [7] Department of Molecular and Cell Biology, University of Connecticut, Storrs, CT, USA. [8] Bell Center for Regenerative Biology and Tissue Engineering, Marine Biological Laboratory, Woods Hole, MA, USA. [9] These authors contributed equally: Hannah Schmidbaur, Akane Kawaguchi. ✉email: calbertin@mbl.edu; elly.tanaka@imp.ac.at; oleg.simakov@univie.ac.at

Cephalopods have the largest invertebrate nervous systems and possess many lineage-specific adaptations such as rapid adaptive camouflage, arms with suckers and camera-type eyes. Many cephalopod characteristics evolved convergently to those of vertebrates, which makes them an attractive system to study the genetic basis of wide-scale organismal innovations and the pathways behind their evolution.

On a genomic level, the emergence of novel genes, extensive gene duplications, and wide-ranging RNA editing have been described in cephalopod genomes[1]. Expansions of gene families such as C2H2s, Protocadherins, and GPCRs, and extensive RNA editing allowed the diversification of protein-coding transcripts in the nervous system and is proposed to have played an important role in its evolution. While similar innovations are known from vertebrate genomes, the mechanisms driving the evolution of these features are different: vertebrates went through several rounds of whole-genome duplications that produced large sets of multi-copy genes and the diversification of their functions, there is no indication for similar events in cephalopods[1–3]. In contrast, it has been suggested that the coleoid cephalopod (squids, cuttlefish, octopus) lineage went through large-scale genome reorganization[2,3].

A property of metazoan genomes is that local gene order or microsynteny is conserved between even distantly related species[4–6]. This conservation is supported by functional studies of regulatory constraints, shown in genomic regulatory blocks (GRBs)[4,5,7], as well as co-expression of neighboring genes in tissues or cell types[8]. Early genome assemblies in several coleoids indicate that local gene order has been greatly disrupted, breaking ancient microsyntenies and bringing previously unlinked genes together[2,3]. This event, potentially at a whole-genome scale, could have affected hundreds of gene families, disrupting the order of genes in comparison to the last common ancestor of coleoid cephalopods and other molluscs. The extent of this event is difficult to estimate due to the lack of chromosomal-scale assemblies in cephalopods. To begin to understand the extent of the genome reorganization and its impact on cephalopod genome biology and evolution, we study the emerging model species *Euprymna scolopes* (Hawaiian bobtail squid). This species has been at the center of symbiosis research for over 30 years[9,10], but is also an attractive model system for evolution and development research due to its small adult size, large egg clutches, and relative ease of culture.

To reconstruct the regulatory landscape in the *E. scolopes* genome, we applied chromosomal conformation capture (Hi-C) and open chromatin profiling techniques (ATAC-seq) as well as collected additional expression data. Hi-C allowed us to both improve the previously published *E. scolopes* genome assembly as well as to capture the three-dimensional organization of the genome. Using comparative genomic approaches, we describe the global nature of the genome reshuffling in coleoid cephalopods and demonstrate the emergence of many microsyntenic regions that were previously unlinked in other species. Our data also reveals interactions between distant genomic loci (the topological organization of the genome) shedding light on the three-dimensional organization of the *E. scolopes* genome, as well as identifying genes located in regulatory loops and topologically associating domains (TADs). Our open chromatin data reveals regions accessible to transcription factors and thus potentially constituting regulatory elements. Together, these data allow us to gain insights into the impact of evolutionary changes in gene linkages and the emergence of novel gene regulation. This study provides the basis for the understanding of the evolution of cephalopod genomes and possible implications on morphological novelties in this clade.

## Results and discussion

**The impact of a large-scale genome reorganization on the coleoid cephalopod genome.** Linkage information from chromosome conformation capture allowed us to reconstruct 46 chromosomal scaffolds in *E. scolopes* ("Methods", Supplementary Notes 1 and 2, Supplementary Fig. 1a, b) based on the published assembly[3]. We then compared the order of genes with orthologs found in another 24 animal species ranging from sponges to vertebrates, which allowed us to reconstruct microsyntenic blocks shared between different clades ("Methods", Supplementary Note 3, Supplementary Fig. 2a, Supplementary Data 1). Briefly, we define microsyntenic blocks as at least three or more co-occurring orthologous genes with up to five intervening genes with no constraints on their collinearity. This definition of microsynteny yields the fewest false-positive blocks (compared to just pairs of genes) while providing enough flexibility to detect syntenic regions that underwent local rearrangement and expansion. We recover 505 microsyntenies unique to cephalopods, representing blocks of genes only found in close proximity to each other in *E. scolopes* and at least one octopus species. For the same species sampling and same microsynteny detection parameters only 2 blocks would have been expected by chance (median from 3 rounds of randomization, as described in[6]). Five out of these 505 blocks were paralogous. In total, only 48 out of 2290 genes in these 505 blocks were identified as orphan genes with no homology outside of cephalopods, while all others have orthologs in other animals, suggesting that the origin of microsynteny was due to changes in gene locations rather than novel gene emergence. These microsyntenies have been conserved in coleoid cephalopods despite their long divergence time (Fig. 1b, c), suggesting an evolutionary constraint that kept those blocks of genes together. Similarly, a comparison in other molluscs, such as the scallop *Mizuhopecten yessoensis*[11] and the bivalve *Crassostrea gigas* showed that a much smaller number of bivalve specific microsyntenies (152) is shared between these species. To infer the set of highly conserved microsyntenic blocks, we reconstructed microsyntenies shared between *E. scolopes* and at least six more distantly related species out of a set of 23 species (Supplementary Fig. 2). We recovered 275 such metazoan microsyntenies, which are retained in the *E. scolopes* lineage and are inferred to date back to at least the last common bilaterian ancestor (Fig. 1c, Supplementary Fig. 2a, "Methods"). In comparison, the bivalve *M. yessoensis* retains a similar number of metazoan microsyntenies (216). These results provide evidence for a large-scale microsyntenic gain in coleoid cephalopods.

**Chromosomal distribution and properties of metazoan and novel cephalopod microsyntenies.** Both cephalopod-specific and conserved metazoan microsyntenic blocks are present on 44 out of 46 chromosomal scaffolds (with two chromosomes being too small to contain any microsyntenies). While some chromosomes have higher proportion of novel cephalopod microsyntenies (Supplementary Fig. 1b, c), both microsynteny types are intermixed in the genome (Fig. 1d). This result suggests a genome-wide mechanism for the emergence of novel microsyntenies. The vast majority of single-copy genes (71%) that comprise 232 novel cephalopod microsyntenies are located on different chromosomes in the scallop *M. yessoensis*. As the organization of the recently published Nautilus genome[12] is similar to other molluscs, these results suggest that either many translocations or chromosomal-level fusions occurred in the coleoid ancestor.

**Genomic properties of novel and conserved microsyntenies.** Novel cephalopod and conserved metazoan microsynteny show different genomic properties. Novel cephalopod microsyntenies are on average smaller in size than the metazoan microsyntenies

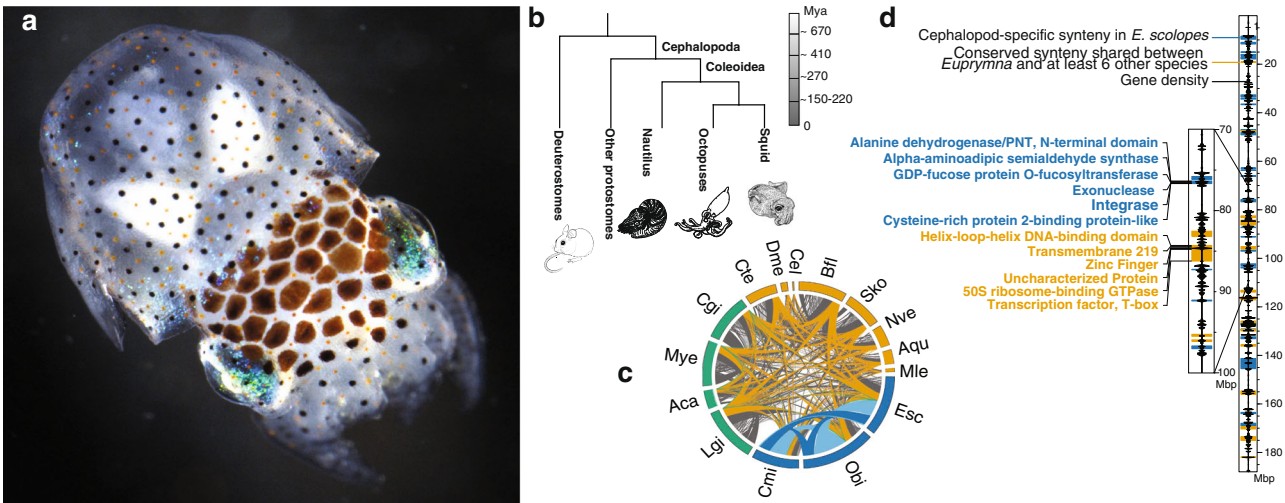

**Fig. 1 Large-scale syntenic reorganization of cephalopod genomes. a** Photograph of an *Euprymna scolopes* hatchling. **b** Schematic tree with divergence times of major cephalopod lineages from deuterostomes and other protostomes[53,54]. **c** Circos plot showing loss of syntenies conserved in other metazoans and the emergence of a large number of novel, cephalopod-specific microsyntenies within cephalopods. Each line represents a syntenic cluster shared between different species. Orange lines indicate syntenic clusters shared between at least seven out of these 24 species (ancestral, metazoan clusters); green lines represent novel molluscan syntenies, shared between five or more molluscs but not present in any non-molluscan species. Blue lines represent cephalopod-specific syntenies shared between either all three cephalopod species (dark blue) or two of the three cephalopod species (light blue) but not present in any non-cephalopod species; gray lines represent other syntenies that do not fall in either of the previous categories. Abbreviations: Aca - *Acanthaster planci*, Aqu – *Amphimedon queenslandica*, Bfl - *Branchiostoma floridae*, Cel - *Caenorhabditis elegans*, Cgi - *Crassostrea gigas*, Cmi - *Callistoctopus minor*, Cte - *Capitella teleta*, Dme - *Drosophila melanogaster*, Esc – *Euprymna scolopes*, Lgi - *Lottia gigantea*, Mle - *Mnemiopsis leidyi*, Mye - *Mizuhopecten yessoensis*, Nve - *Nematostella vectensis*, Obi – *Octopus bimaculoides*, Sko - *Saccoglossus kowalevskii* **d** Example of one whole chromosomal-scale scaffold (right) of *E. scolopes* showing the distribution of gene density, cephalopod-specific (blue) and conserved, metazoan (orange) syntenies. Inset (left) with locations of genes within two specific syntenic blocks.

still present in cephalopod genomes (Supplementary Fig. 2b), despite having similar numbers of genes (Supplementary Fig. 2c). While introns of genes in cephalopod-specific microsyntenies are smaller than those of metazoan microsyntenies, the majority of size differences stem from intergenic regions (~0.2 kb compared to ~7 kb difference, respectively).

We also find evidence for differential enrichment of functional categories between the two microsynteny types. Metazoan microsyntenies[6] are enriched in signaling pathway components of the Wnt-signaling pathway, neurotransmitter transport and synaptic vesicle exocytosis, G-protein coupled receptor signaling, negative regulation of transcription, and BMP signaling pathways, among others. Genes in novel, cephalopod-specific microsynteny, on the other hand, play a role in translation, redox processes, regulation of store-operated calcium entry, mRNA cleavage, transport, and chromatin organization (*p*-values <0.05) (Supplementary Fig. 3).

**Spatial organization of the *E. scolopes* genome**. Three-dimensional chromatin structure including topologically associated domains (TADs) facilitates distant regulatory interactions involved in gene regulation[13,14]. While very little data exists to-date on invertebrate genome topological organization, we found that in comparison to the data known for vertebrate TAD sizes the interaction distances were generally larger in the squid (Fig. 2a, b, Supplementary Note 4). TAD prediction tools (see "Methods") reveal a median *E. scolopes* TAD size of 2.5 Mb, in comparison to an average of 1.2 Mb in human[15]. In addition, the distribution of TAD sizes in *E. scolopes* was considerably wider than in human, suggesting a higher variability.

In vertebrates, TAD formation is mediated by proteins, such as CTCF and cohesin[16,17]. We infer that the same mechanisms, including CTCF and the proteins Smc1 and Smc3 of the cohesin complex, are present in the *E. scolopes* genome and conserved with other animals, suggesting that similar mechanisms may be

deployed in cephalopods (Supplementary Fig. 4). We also find TAD boundaries to be enriched for a CCCTC-like motif[18–20] reminiscent of a CTCF binding site (Fig. 2c, *p* = 1e−12, "Methods", Supplementary Note 4).

**Topological organization around microsyntenies**. Several studies suggested the possibility of correspondence between micro-synteny and regulatory domains in metazoan genomes[4,5,7,8,21–23]. To understand the relationship of microsyntenies and TADs we compared the localization of randomly computed microsyntenic blocks, that follow the same properties as our observed blocks but are randomly distributed throughout the genome to the observed microsyntenies ("Methods"). We find a tendency of conserved metazoan microsyntenies to be localized towards the center of the predicted TADs, whereas new cephalopod microsyntenies appear to be more evenly distributed (Fig. 3a).

To further study the relationships of genomic regions and their interactions independent of TAD predictions, we computed a tree structure reflecting the organization of each chromosomal scaffold ("Methods", Supplementary Note 4). Each bifurcating branch reflects the relationships of genomic regions in Hi-C signal strength, allowing us to track interaction intensities in microsyntenies (Fig. 3b). Surprisingly, we found that novel microsyntenies are more likely to form tight interaction regions, reflected by subtrees with few branches, when compared to randomly sampled syntenies (Fig. 3b). This result indicates significantly higher levels of compartmentalization in both cephalopod and ancestral microsyntenies.

Motivated by the importance of three-dimensional genome architecture and microsyntenic co-localization in *Euprymna scolopes*, three-dimensional modeling based on Hi-C interaction matrices was performed ("Methods", Supplementary Fig. 5, Supplementary Note 5) to provide a deeper understanding of spatial properties and co-localization of both novel and ancient

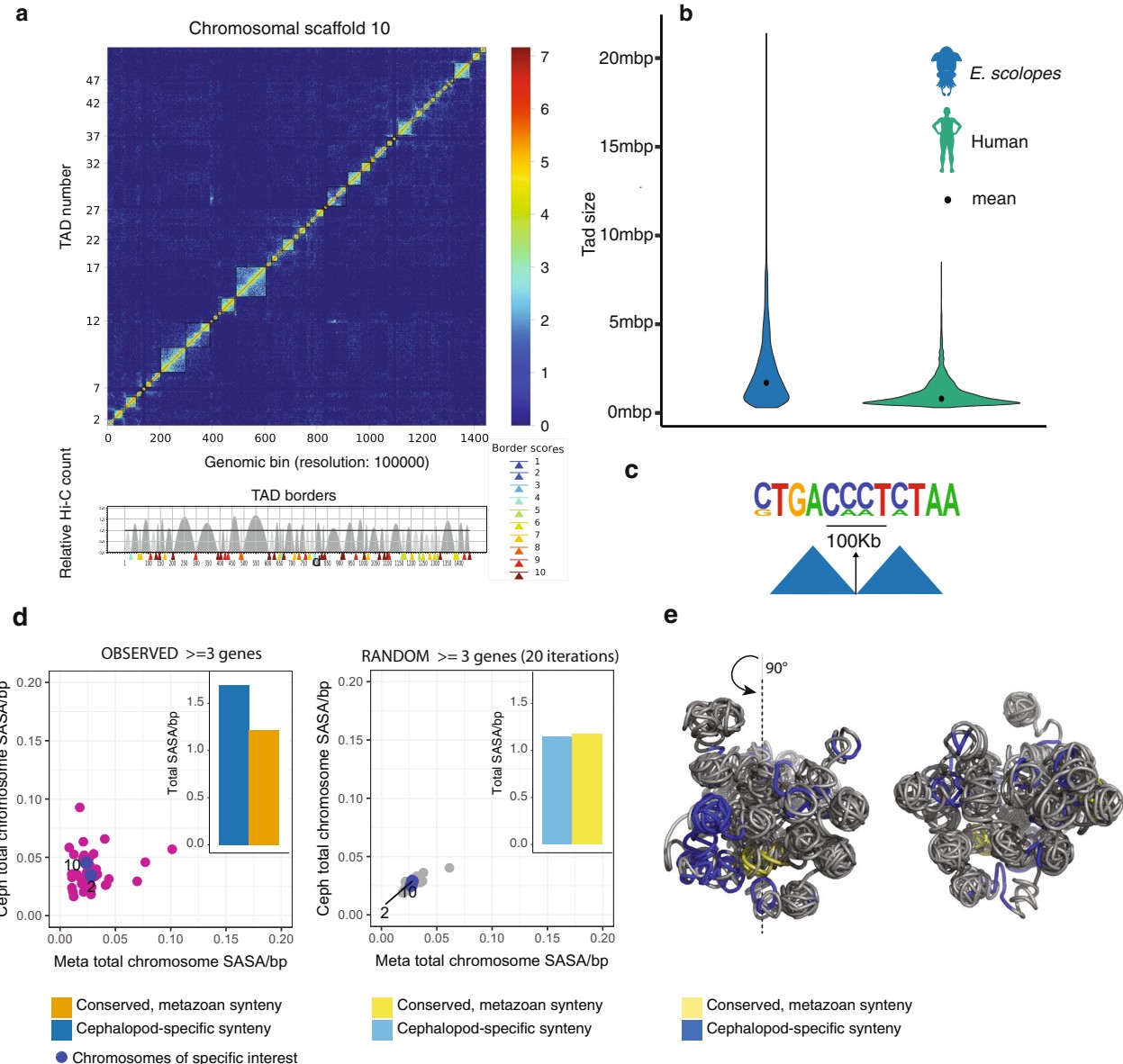

**Fig. 2 Topological and spatial organization of the cephalopod genome. a** Top: Hi-C normalized interaction matrix at 100 kb resolution for chromosomal scaffold 10. Bottom: Relative Hi-C count and TAD boundary scores as predicted by tadbit (1 = lowest, 10 = highest). **b** Violin plot of TAD size distributions for human and *Euprymna scolopes* computed at 100 kb resolution, plotted in 100 kb bins. **c** CTCF binding motif as identified by homer motif search in 100 kbp of predicted TAD boundaries. **d** Solvent area surface exposure (SASA) per bp for individual chromosomal scaffolds (observed and random). Conserved metazoan synteny on x-axis, cephalopod-specific microsynteny on y-axis. **e** Three-dimensional model of chromosomal scaffold 10 with novel (blue) and conserved (yellow) microsynteny locations labeled. Left and right models are the same, shifted by 90°.

microsyntenic regions within modeled chromosomal scaffolds. Three-dimensional models revealed that novel cephalopod microsyntenies have distinct spatial properties from ancient microsyntenies. In particular, both synteny types showed differential solvent accessibility on some chromosomes when compared to random distributions (Fig. 2d). Moreover, novel cephalopod microsyntenies were on average less buried, thus covering a larger proportion of chromosomal surface (Supplementary Fig. 6). This result was in contrast to the conserved metazoan microsyntenies, which tend to participate in the formation of the chromosomal core (Fig. 2e, Supplementary Fig. 6). Since the novel microsyntenies are transcriptionally active (Fig. 3, see below), their location on the chromosomal surface may be reflective of highly dynamic inter-chromosomal regulation, as well as being more accessible to transcription factors.

The GC content of metazoan and cephalopod-specific microsynteny was evaluated along with predictions of A/B compartments based on the Hi-C interaction matrix ("Methods"). The analysis did not provide sufficient evidence for one of the mycrosyntenic types being more prevalent in either of the compartments. Until further experimental data are available (such as methylation and acetylation profiling) for *Euprymna scolopes*, we cannot accurately infer the distribution of cephalopod-specific and metazoan microsynteny within A/B compartments.

Taken together, the strong genomic conservation among sequenced cephalopods, the comparably tight packaging (short inter-gene distances) of microsyntenic clusters and their prevalent association with defined subcompartments within detected TADs suggests strong selective pressure to maintain regulatory properties of novel microsyntenic units in the cephalopod genomes.

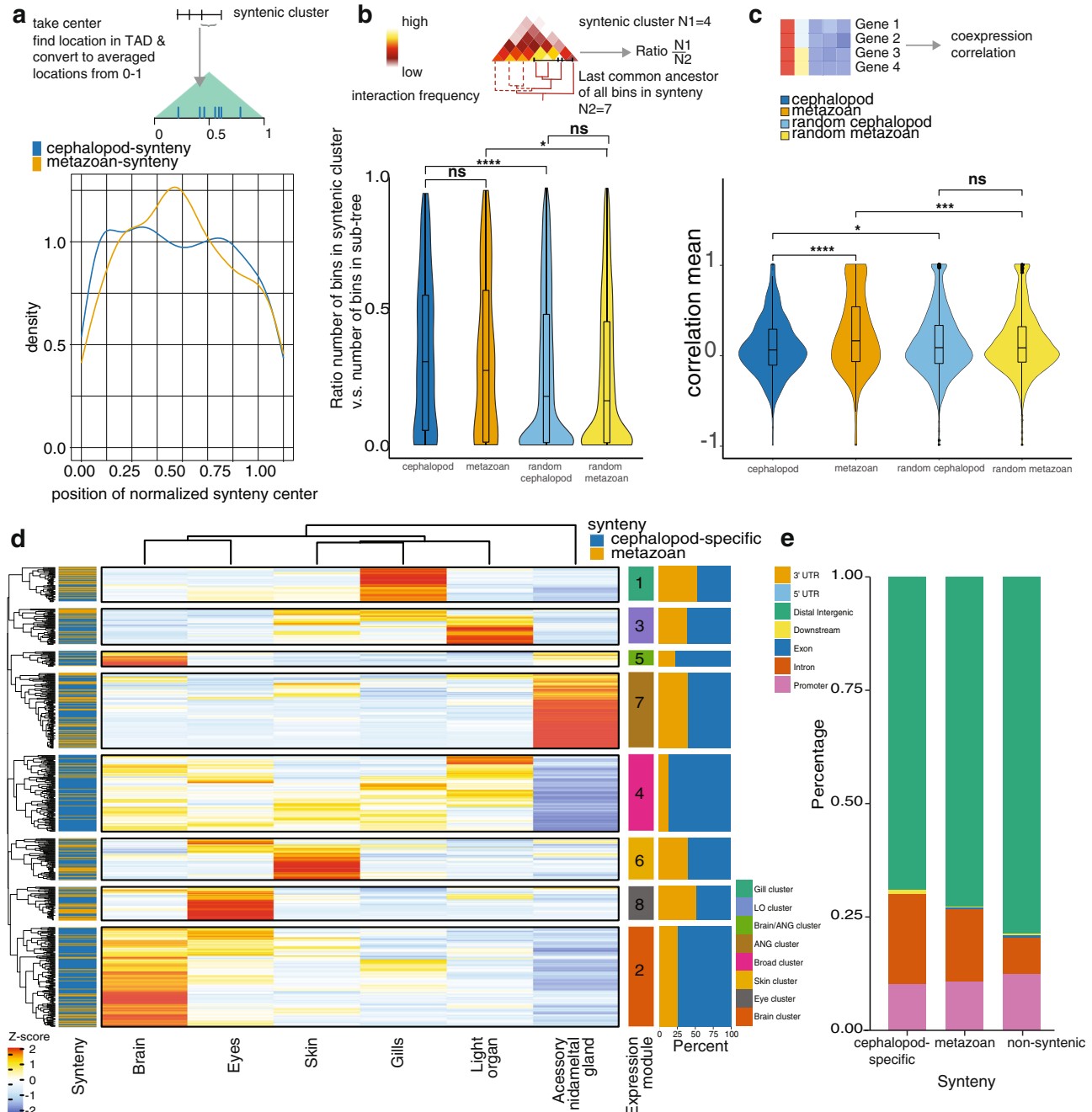

**Co-expression properties of microsyntenies.** Co-expression of syntenic genes is an important property that can reflect their regulation. Genes in cephalopod-specific microsyntenies do not tend to be co-expressed, despite their tight co-localization ("Methods", Supplementary Note 7). When compared to randomly sampled groups of genes that follow a similar distribution to the observed, novel microsyntenies, the mean co-expression coefficient is even slightly lower in the observed data (Wilcoxon test, $p \leq 0.05$, Fig. 3c). In contrast, conserved metazoan microsyntenies show significant (Wilcoxon test, $p \leq 0.001$) co-expression when compared to simulated microsyntenies (Fig. 3c). This result indicates that genes in metazoan microsynteny tend to co-express in a defined set of tissues, similar to previous findings for ancient metazoan microsynteny[8]. A similar pattern was also observed for co-expression of novel and conserved metazoan microsyntenies in *O. bimaculoides* (Supplementary Fig. 7a). No type of microsynteny showed enrichment of expression-specificity in a particular tissue[24].

To further categorize expression profiles, we investigated averaged expression of syntenic regions. This analysis showed complex patterns that fell into eight distinct expression modules across adult *E. scolopes* tissues (Fig. 3d, "Methods", Supplementary Note 7). Although most modules had similar proportions of both microsynteny types, some clusters formed outliers. For example, module 8 showed eye-specific expression and was mostly dominated by metazoan microsynteny. Interestingly, clusters encompassing multiple nervous tissues, in particular modules 2 and 4, were enriched in novel cephalopod microsyntenies (Fisher's exact test p-values $\leq 0.02$ and $\leq 1e{-}07$, respectively). Their orthologs were similarly expressed in *O. bimaculoides* nervous tissues, with novel microsyntenies dominating in modules associated with the strongest brain expression (Supplementary Fig. 7b). However, the overall module correspondence was impacted by the difference in tissue sampling (Supplementary Fig. 7c).

**Fig. 3 Novel cephalopod microsyntenies and their regulatory properties in *Euprymna scolopes*. a** Ratio between densities of the center of observed and randomized microsynteny locations within normalized TADs. Ratio increases towards the center of TADs for metazoan syntenies and decreases for cephalopod synteny. **b** Compactness of novel and metazoan microsynteny, compared to random simulations. Microsyntenic clusters must contain at least 3 genes, if fewer genes were annotated to the tree, these clusters were excluded. Plotted are the ratios between the number of bins (at 40 kb Hi-C resolution, cephalopod micrysynteny bins $n = 143,969$, metazoan microsynteny bins $n = 82,417$, random cephalopod microsynteny bins $n = 3,499,849$, random metazoan microsynteny bins $n = 1,889,687$) in microsyntenic clusters (within 7 and 25 bins, valid microsyntenies: cephalopod $n = 265$, metazoan $n = 125$, random cephalopod $n = 4265$, random metazoan $n = 2180$) and the number of "descendant" bins from the last common ancestor of those microsyntenic bins ("Methods", cephalopod $n = 143,969$, metazoan $n = 82,417$, random cephalopod $n = 3,499,849$, random metazoan $n = 1,889,687$). The ratio ($n = 6835$) of the number of bins in a cluster and the number of bins in the sub-tree were compared by two sided Wilcoxon rank-sum test comparing the linked groups (****$p < 0.0001$, * $< 0.05$, ns: not significant). The closer the ratio to 1, the lower is the difference between the size of the syntenic block and the number of bins in an extracted tree. Violin plots—distribution, boxes—interquartile range (cephalopod = lower 0.06, upper 0.56, metazoan = lower 0.01, upper 0.58, random cephalopod = lower 0.01, upper 0.49, random metazoan = lower 0.01, upper 0.46), bars—median (cephalopod = 0.31, metazoan = 0.28, random cephalopod = 0.18, random metazoan = 0.17), whiskers—furthest sample within 1.5x interquartile range (cephalopod = min 0.002, max 0.94, metazoan = min 0.002, max 0.95, random cephalopod = min 0.002, max 0.96, random metazoan = min 0.002, max 0.96), maximum and minimum ratios: cephalopod = min 0.00189, max = 0.941, metazoan = min 0.00217, max = 0.952, random cephalopod = min 0.00151, max 0.962, random metazoan = min 0.00150, max 0.96. **c** Co-expression correlation of genes in microsyntenic clusters (cephalopod $n = 476$, metazoan $n = 236$, random cephalopod $n = 6925$, random metazoan $n = 4038$). The co-expression correlation of metazoan syntenies is higher than that of cephalopod-specific syntenies or random clusters (****$p < 0.0001$, ***$p < 0.001$, * $< 0.05$). Violin plots—distribution, boxes—furthest sample within 1.5x interquartile range (cephalopod = min −0.69, max 0.87, metazoan = min −0.63, max 1.0, random cephalopod = min −0.72, max 0.95, random metazoan min −0.66, max 0.9), bars—median (cephalopod = 0.05, metazoan = 0.15, random cephalopod = 0.07670835, random metazoan = 0.07) outliers were excluded from these numbers. Maximum and minimum values: −1 and 1 in all cases. **d** Clustering of mean expression per synteny cluster, color-coded by synteny type, expression among *Euprymna scolopes* adult tissues. Syntenic clusters form eight expression modules with specific expression patterns. Expression matrix is z-score normalized. Light organ—*E. scolopes*-specific organ harboring symbionts, accessory nidamental gland—female-specific reproductive organ of some squid species. **e** Annotation of ATAC peak location in late organogenesis. Peaks annotated as associated with cephalopod-specific microsyntenies are more often found in intronic regions. Promotor defined as +10 kb and −10 kb predicted transcription start site.

We next wanted to investigate whether these expression patterns are biased due to a single highly expressed gene per syntenic block. The vast majority of microsyntenic blocks (76%, Supplementary Fig. 8b) have one gene that contributes to more than 50% of the cumulative expression. We then calculated relative expression levels across tissues per gene and averaged it for each block, showing that overall expression module identities are retained (Supplementary Fig. 8a). In general, expression variance correlates with absolute expression of genes (Supplementary Fig. 8c). In some metazoan syntenies, however, especially in the tissues defining a module, the variance was low, indicating higher co-expression constraints (Supplementary Fig. 8c).

Together, these results highlight the complex expression domain contribution of microsyntenic regions and identifies a discrepancy in the co-expression dynamics between novel cephalopod and metazoan microsyntenies. This paucity of co-expression in cephalopod microsyntenies indicates a potentially different mode of their gene regulation.

**Regulatory signatures of microsyntenies**. Expression modules showed specific signatures of regulatory motifs associated with them. We predicted regulatory regions using assay for transposase-accessible chromatin (ATAC-seq[25]) data from a developmental time course ("Methods", Supplementary Note 9). Predicted peaks associated with each expression cluster were then further analyzed for known transcription factor motif enrichment, separately for cephalopod and ancient microsynteny ("Methods", Supplementary Data 2). We find that the cephalopod microsynteny module 2, which is associated with multiple nervous tissues, was enriched for the transcription factors binding motifs Chop[26], E2F1[27], NeuroG2[28], COUP-TFII[29], Atf4[30] involved in nervous system differentiation and developmental transcription factors ZBTB18, Esrrb, Tcf21, Pitx1, GATA in all three developmental stages ($p < 1e{-}3$). Module 4 was similarly enriched in transcription factor binding motifs involved in nervous system and general development such as Tcf3, TCFL2, GATA, Tcf21, and Pitx1 in all three developmental stages ($p < 1e{-}3$). This suggests a common regulatory scheme responsible for

gene expression in each of the expression modules and an association of those motifs with novel cephalopod microsynteny.

To complement our ATAC-seq data, we conducted whole-genome alignments between available cephalopod genomes using different alignment similarity and length thresholds ("Methods", Supplementary Note 8, Supplementary Table 5, and Supplementary Fig. 9). This helped identify potential conserved non-coding elements (CNEs) and their association with gene bodies and genome topology ("Methods", Supplementary Fig. 10, Supplementary Note 8). Due to evolutionary distance between squid and octopus lineages, our approach yielded only 1187 coleoid cephalopod CNE candidates with a similarity threshold of 0,95 and a minimum size of 100 bp (Supplementary Table 5), of which 613 could be localized to gene features (Supplementary Fig. 10a–c). 139 were associated with novel cephalopod micro-syntenies (inside or within 1 kb of microsynteny), 73 with metazoan syntenies inside or within 1 kb of microsynteny), and 401 were located outside any synteny (Supplementary Fig. 10b). Only 12 of the 1187 candidates had overlap with ATAC-seq peaks (Supplementary Table 5). For CNEs shared among squid genomes we found 42920 putative CNEs with a similarity threshold of 0.95 and a minimum size of 100 bp (Supplementary Table 5), of which 13889 could be localized to gene features (Supplementary Fig. 10a–c). 2444 were associated with novel cephalopod microsynteny (inside or within 1 kb of microsynteny), 3255 with metazoan synteny, and 8190 were located around genes outside any synteny (Supplementary Fig. 10c). Similarly, very few overlapped with ATAC peaks (14). Therefore, the regulatory role of these regions remains unclear.

A potential contribution to this observation could be the high evolutionary turnover of regulatory regions within the squid lineage, diminishing the insight gained from genome alignment/conservation-based inferences. Cephalopod genomes are large with over 50% of genome length attributed to repetitive elements[2,3]. We thus assessed the transposable element composition of ATAC-seq peaks associated with microsyntenic clusters. Both microsynteny types showed an average repeat content of 40%. ATAC-seq peak sequences in cephalopod microsyntenies in

*E. scolopes* showed an elevated repetitive element content of 82% and were most frequently associated with LINE/CR1-Zenon elements (44%, compared to 35% in metazoan microsyntenic peaks). The LINE/CR1 expansion was identified as the most common and specifically expanded repeat element class in the squid (*E. scolopes*) lineage[3].

**Evolutionary scenarios for functional microsynteny emergence in cephalopods**. Conserved gene co-localization may be explained by regulatory regions within a neighboring gene, even though the function of the gene may be unrelated (the bystander scenario), together forming a genomic regulatory block (GRB)[4,5], or via shared regulatory elements controlling the expression of all syntenic genes, resulting in higher co-expression[8]. Our microsynteny approach is agnostic to this distinction, focusing on any conserved co-localization of three or more genes. We thus may use our data to independently profile the propensity of metazoan or cephalopod microsyntenies to correspond to known functional microsyntenic models. The location of ATAC-seq peaks within the microsyntenic clusters reveals that open chromatin regions associated with cephalopod-specific microsynteny were more often found in introns than peaks found in conserved, metazoan microsynteny or other non-syntenic genes (Fig. 3e, Fisher's exact test $p < 1e−5$). Interestingly, while sharing little overlap, CNE distribution, in particular of CNEs shared among squids, showed significant localization towards distal regions in metazoan syntenies (Fisher's exact test $p < 2.2e−16$) (Supplementary Fig. 10). This result indicates that, unlike more conserved microsyntenies, novel cephalopod microsyntenies are more similar to the GRB scenario, in particular the bystander model, in which putative enhancers are found within closely located (bystander) genes inside the same microsyntenic cluster and may potentially lack distal regulatory domains. This observation may also complement the finding of weaker co-expression of microsyntenic genes in cepholopod-specific microsynteny, compared to more ancient, conserved microsyntenies.

Given this insight, we further sought to investigate the potential function of novel cephalopod microsyntenies in expression modules 2 and 4 that showed the highest contribution to neuronal tissue expression domains. Such microsyntenies can be considered a useful test set of functional microsyntenies that were involved in the evolution of the cephalopod nervous system. We thus examined the genomic rearrangement, the regulatory landscape, and expression of genes within one of the representative cephalopod-specific microsyntenies from expression module 2. It was one of the clusters with the highest number of genes, encoding for ceramide-1 phosphate transfer protein, phenylalanine-tRNA ligase, splicing factor 3B subunit, integrator complex subunit, and amyloid protein-binding protein. Orthologs of this microsynteny were widely spread across two chromosomes in scallop (Fig. 4a), yet were densely packed in the *E. scolopes* genome with almost no intergenic space and a few dominant ATAC peaks towards one end of the cluster in the intron of phenylalanine-tRNA ligase (Fig. 4b). Similar to the general trend, this clustering in cephalopods implies either several local translocations or a large-scale chromosomal fusion, followed by rearrangements. The cluster was localized towards the center of the predicted TAD (Fig. 4b), close to another novel microsyntenic unit (from the same expression module). Together, these two units form a compartment of high Hi-C interaction density, separate from the closest metazoan microsyntenic compartment and its associated ATAC-peaks (Fig. 4b). In our three-dimensional chromosomal model, this microsynteny was also found on the surface of chromosomal scaffold 2 (Fig. 4c). Interestingly, the genes in this cluster show nervous system expression both in scallop and *E. scolopes* (Fig. 5a, Supplementary Note 10). Despite some of these

genes being considered as purely metabolic or housekeeping genes, in vertebrates they are known to play an important role in nervous system development and activity[31–35]. We conducted in situ hybridization to visualize gene expression during development in *E. scolopes* embryos and confirmed the expression in all major central brain regions (Fig. 5b, c, Supplementary Note 11). However, we also revealed expression in other novel cephalopod tissues and organs, such as the axial nerve cords as well as heart and gills (Fig. 5b, c, Supplementary Fig. 11). This result provides evidence that this microsyntenic cluster and its siblings from expression module 2 could comprise functional microsynteny of the bystander model type that were crucial contributors to the emergence of novel cephalopod expression domains. More generally, the observation of microsyntenies associated with cephalopod innovations (MACIs) and their further investigation could help dissect the evolution of complex cephalopod tissue expression patterns.

In summary, we present a comprehensive study of topological and regulatory genome organization in a coleoid cephalopod. Characterized by megabase-range interactions, cephalopod genomes have been impacted by a genome-wide syntenic reorganization, with an extent that is rare among animal genomes. This reorganization led to the gain of hundreds of cephalopod-specific microsyntenies that are associated with compact topological regions and a distinct mode of gene regulation. Their putative regulatory sequences were often located within the introns of genes within the same microsyntenic cluster, as has been proposed for functional gene linkage in the bystander model. Our analysis of the microsyntenic expression data revealed complex expression patterns of novel microsyntenies associated with a specific set of cephalopod neural tissues and other novel organs. We identify two such expression modules most prominently affected by the emergence of novel cephalopod microsyntenies, each associated with a specific regulatory signature. We propose that this syntenic 'locking-in', i.e., high compactness and regulatory streamlining, was responsible for the emergence and extension of ancestral molluscan neural tissue expression domains. As much of cephalopod molecular biology remains elusive, our study proposes the use of these microsyntenies associated with cephalopod innovations (MACIs) to begin to unravel molecular changes associated with cephalopod developmental and organismal innovations. This study sets the stage for further investigation of MACIs and their roles in the emergence of novel expression domains and organismal innovations in cephalopods.

## Methods

**Collection of animals**. *E. scolopes* eggs were obtained from cultures and maintained at the Vienna Zoo or at the Marine Biological Laboratory. All work was performed in compliance with the EU Directive 2010/63/EU on cephalopod use and AAALAC guidelines on the care and welfare of cephalopods[36]. Adult *E. scolopes* spawned naturally in their tanks, and embryos were collected shortly after spawning and maintained in a closed aquarium system filled with artificial seawater. Embryos developed to the appropriate stage and were anesthetized with 2% Ethanol before use[37–39].

**Hi-C, genome scaffolding, and 3D analysis**. Hi-C sample preparation was performed as described in Supplementary Note 2. Briefly, Hi-C samples were generated at developmental stage 27[40] with 30 pooled embryos using the six base restriction enzyme Hind3. Paired-end sequencing of 50 bp was done on an Illumina HiSeq2500. Hi-C reads were aligned to the reference genome (excluding scaffolds <50k) resulting in over 106 million valid interaction pairs (alignment rate ~71%). Aligned reads were used to scaffold the genome to chromosomal scaffolds. Assembly statistics are summarized in Supplementary Note 2. Raw Hi-C reads were then again mapped to the new chromosomal scaffolds, recovering over 106 million valid interaction pairs (alignment rate ~71%). Three-dimensional modeling of chromosomal scaffolds is described in Supplementary Note 3. For human samples, Hi-C samples of B-lymphoblastoids were downloaded from NCBI (ref. [41], SRR1658570, HIC001) and aligned to the human reference genome (GRCh38.p12) obtaining over 144 million valid interaction pairs (~73% alignment rate).

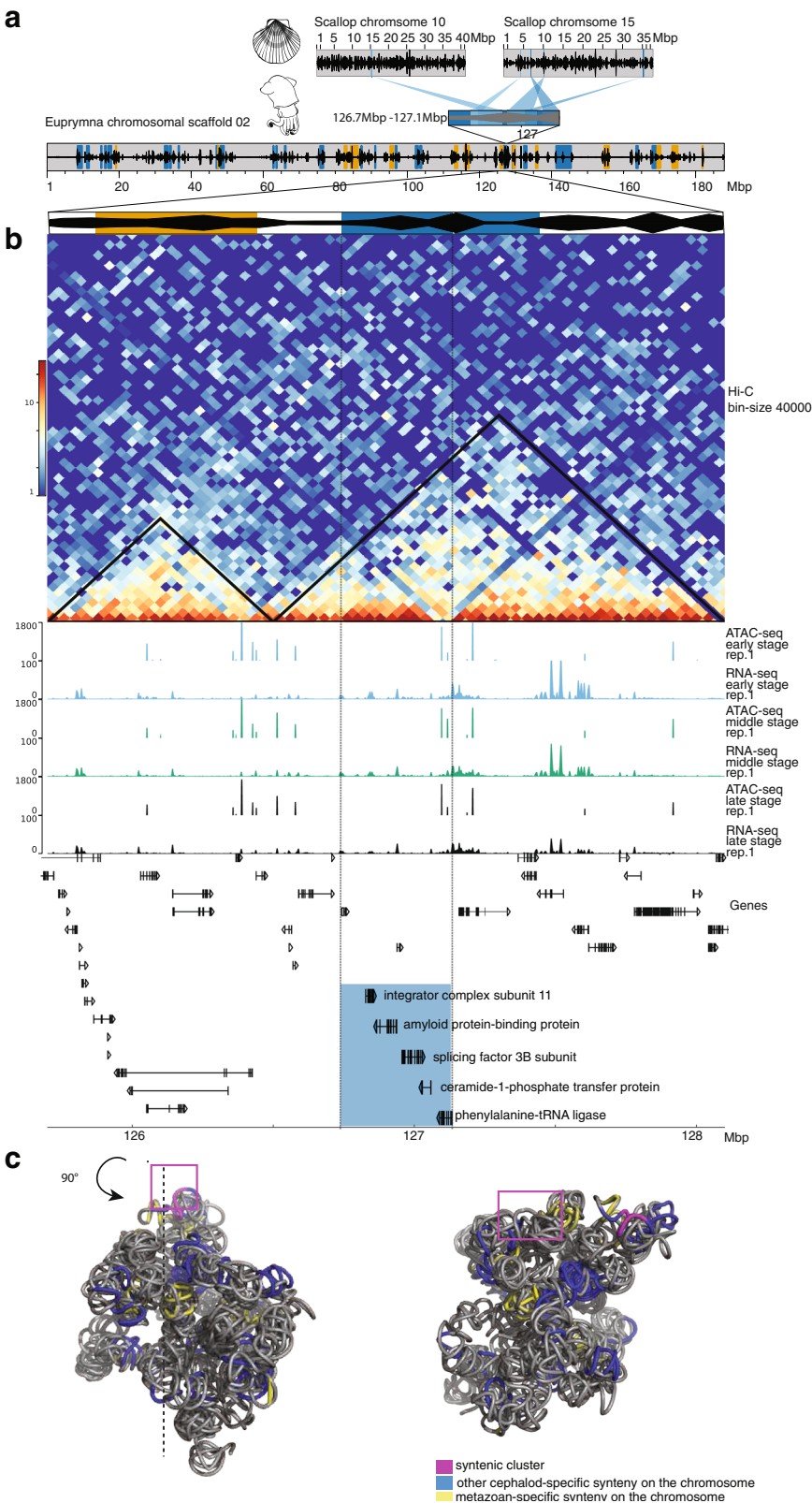

**Synteny analysis**. Gene orthology was reconstructed using 27 species spanning all major metazoan clades. Microsynteny was computed using in-house tools as described in detail in Supplementary Note 3. Metazoan synteny was defined as all syntenic blocks shared between at least seven other species. Novel, cephalopod-specific synteny was defined as synteny shared between *E. scolopes* and at least one octopus species. Random microsyntenies were modeled after the distribution of observed syntenies as described in ref. [8] for 20 iterations. Additional details of the scripts and steps are found in Supplementary Note 3.

**Chromatin conformation analysis**. TADs for *E. scolopes* and Human were called with Tadbit[42] with the Tadbit algorithm and with HiCExplorer[43]. *E. scolopes* TADs were averaged and the location of the middle of each syntenic cluster was mapped to analyse the distribution of syntenies within TADs. If syntenies spanned several TADs, only the TAD mapping the middle of that synteny was considered. To analyse the topology of microsyntenic clusters further, the normalized Hi-C interaction matrix was used to cluster each bin to its closest neighbor by the bin interaction strength. An interaction cladogram for each chromosome was

**Fig. 4 Emergence of compact clusters and their unique expression patterns in nervous tissues. a** Location of orthologous genes of one of the MACIs in *Mizuhopecten yessoensis*. The genes are located on two separate chromosomes (top—whole chromosome, bottom—zoom in, black—gene density, blue—location of orthologous genes) with many intervening genes in-between. Scallop orthologs of genes in the microsyntenic cluster are plotted at the bottom highlighted in blue. **b** Genes of the same cluster in *Euprymna scolopes*. The genes (highlighted in blue) are tightly packed on one chromosomal scaffold with only one intervening gene (top—whole chromosomal scaffold, bottom—zoom in, orange—conserved, metazoan microsyntenic clusters, blue—cephalopod-specific microsyntenic clusters). The cluster is located within a TAD. Two major ATAC-seq peaks are present in intronic regions of the last gene of the cluster in three developmental stages (y-axis—signal value). RNA-seq read count of three developmental stages shows several small peaks in the region of the cluster (y-axis—read count). Early-stage—early organogenesis, stage 20, middle stage—late organogenesis, stage 24/25, late stage—close to hatching, stage 28/29. **c** Three-dimensional reconstruction of chromosomal scaffold 2. The cephalopod-specific syntenic cluster is located on the surface. Left and right views are shifted by 90°.

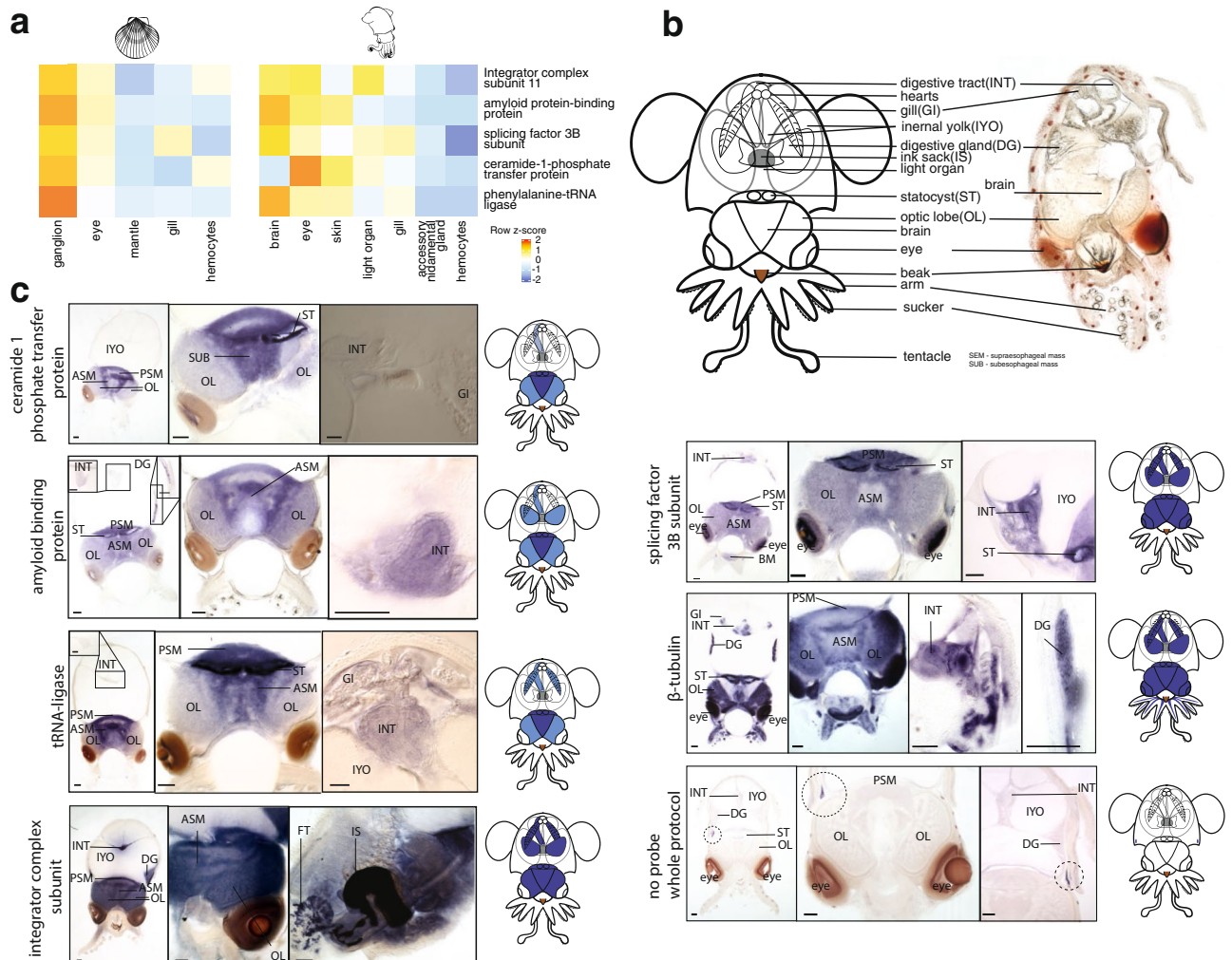

**Fig. 5 Expression of genes in a representative MACI in late developmental stages of *E. scolopes*. a** Heatmap showing expression of orthologous genes of the cephalopod-specific cluster in adult scallop and *E. scolopes* tissues showing neuronal expression in both species. Scale bar shows z-score normalized expression levels per row. **b** Scheme of late stage (Stage 27–29) *E. scolopes* anatomy and section of hatchling. **c** Expression of MACI genes in nervous tissues and inner organs of late developmental stages of *E. scolopes*. All genes show expression in different lobes of the brain, most dominantly the ASM and PSM. Light expression patterns are present in the optic lobes, the digestive gland and intestines in most genes. Expression patterns differ to Beta-tubulin, which was chosen as a control for its pan-neuronal expression domain, in its intensity and distribution. Scale bars = 100 μm. ASM anterior subesophageal mass, DG digestive gland, ES esophagus, FT funnel organ, GI gills, INT intestine, IS ink sack, IYO internal yolk, OL optic lobe, PSM posterior subesophageal mass, ST statocyst. Dotted circles - color trapping.

reconstructed that way. To understand how well a syntenic region is defined by its interactions, we extracted the last common ancestor of that region (i.e., the bins in that region) from the whole tree making up the chromosome. Then the ratio between those sub-trees and the number of bins in a syntenic cluster was calculated and differences between groups were tested using the Wilcoxon test.

**Chromatin conformation analysis of synteny.** The center of each syntenic cluster was localized within predicted TAD boundaries. The locations were then normalized and plotted for observed and random microsyntenies. To visualize the differences between observed and random, the ratio between densities of each were calculated and plotted in normalized TAD locations (Fig. 3a). A density over one signifies an enrichment of observed synteny compared to random clusters. A density lower than one signifies a depleted signal of observed synteny compared to random clusters. Random synteny represents clusters sampled from the distribution of observed blocks from random locations in the genome.

**Co-expression analysis**. Co-expression coefficients for adult tissues of *E. scolopes* and *O. bimaculoides* were computed as described in ref. [8]. A detailed description of the steps is provided in Supplementary Note 7.

**CNE identification**. The genomes of *E. scolopes* and *O. bimaculoides* and of *E. scolopes* and *A. dux* were aligned with megablast, using *E. scolopes* as the query sequence. Five different settings for BLAST[44] similarity scores (-perc_identity) were used: 0%, 70%, 80%, 95%, and 98% (Supplementary Figs. 9, 10, Supplementary Table 5, see refs. [4,7,45,46]). For further settings see Supplementary Note 8. Multi-mapping regions were excluded if they overlapped by more than 50% and occurred more than 3 times using BEDOPS[47] bedmap. `bedmap --count --echo --fraction-both 0.5 --delim '\t' prefiltered_megablast.bed | awk '$1<4' | cut -f2- |sort-bed - | uniq`. Remaining overlapping regions were merged with `bedops –merge`. Any region overlapping with an exon by 1 bp or more was excluded using bedtools[48] subtract. To exclude repetitive regions, fasta sequences were extracted from the filtered putative CNE locations and meme's[49] dust (cut-off 10) function was used to mask repeats. In addition, two datasets were created for each similarity score with a minimum size of 100 bp or 50 bp respectively. For similarity scores of 0%, only 100 bp regions were kept. Any region with more than 25% Ns was excluded. To remove any remaining coding sequences, the remaining putative CNE sequences were blasted against the NCBI[50] NR database and any regions overlapping with a BLAST match were removed.

**Chromatin accessibility assay with ATAC-seq**. ATAC-seq samples preparation and analysis are described in Supplementary Note 9. ATAC-seq was generated for stages 20, 25, and 28/29[40] with two biological replicates each as described in refs. [25,51,52] with slight modifications. Each ATAC-seq library was generated with two biological replicate samples. Samples were sequenced on Illumina HiSeqV4 using 125 bp paired-end reads. The reads were trimmed with BBDuk (https://jgi.doe.gov/data-and-tools/bbtools/bb-tools-user-guide/bbduk-guide/) and mapped to the chromosomal scaffolds. Peaks were called with Genrich (https://github.com/jsh58/Genrich). After trimming between 72 and 143 million reads remained which were mapped at between 79 and 83% and between 22,443 and 36,933 peaks were called for each sample.

**Fluorescence in situ hybridization (FISH) and in situ hybridization (ISH)**. *E. scolopes* embryos removed from eggs and jelly layers and hatchlings were anesthetized in 4% EtOH in seawater or 4% EtOH and MgCl$_2$ (2 M solution added slowly to seawater) and subsequently fixed in 4% paraformaldehyde[38]. Sequences of interest were identified from the adult *E. scolopes* transcriptomes. cDNA of pooled developmental stages was used for PCR with Q5 polymerase. Products were cloned in pjet vectors and isolated with an innuPREP Plasmid Mini Kit (Analytik jena (Jena, Germany)) and sequenced. Riboprobes were generated from amplified minipreps and reverse transcribed with DIG-labeled nucleotides. Details on the FISH and ISH protocol can be found in Supplementary Note 11. Embryos and hatchlings were imaged on an inverted Zeiss (Oberkochen, Germany) LSM 780 multiphoton Confocal Laser Scanning Microscope.

**Reporting summary**. Further information on research design is available in the Nature Research Reporting Summary linked to this article.

## Data availability

The data that support this study are available from the corresponding authors upon reasonable request. The Hi-C and ATAC-seq data have been deposited in the NCBI database under Bioproject PRJNA661684. All expression, ATAC-seq, and CNE data that is mapped to the reference genome is available on a genome browser (currently, http://metazoa.csb.univie.ac.at:8000/euprymna/jbrowse or upon request). All other genomic and trascriptomic data used was downloaded from NCBI (GCA_002113885.2, GCA_000002075.2, GRCh38.p12, GCA_001949145.1 OLI-Apl_1.0, GCA_000003605.1, GCA_000224145.2, GCA_000003815.1 Version 2, GCA_004765925.1, Spur_3.1, GRCm38.p6, SAMN00691532, SAMN00152410), ENSEMBL (BDGP6.28 http://www.ensembl.org/Drosophila_melanogaster/Info/Index, WBcel235 http://m.ensembl.org/Caenorhabditis_elegans/Info/Annotation, Capitella_teleta_v1.0 http://metazoa.ensembl.org/Capitella_teleta/Info/Index, ASM23792v2 http://metazoa.ensembl.org/Schistosoma_mansoni/Info/Index, oyster_v9 http://metazoa.ensembl.org/Crassostrea_gigas/Info/Index, Helro1 http://metazoa.ensembl.org/Helobdella_robusta/Info/Index, Lotgi1 http://metazoa.ensembl.org/Lottia_gigantea/Info/Index, PRJNA270931 https://metazoa.ensembl.org/Octopus_bimaculoides/Info/Index, Stegodyphus_mimosarum_v1 [https://metazoa.ensembl.org/Stegodyphus_mimosarum/Info/Index], Tcas5.2 [http://metazoa.ensembl.org/Tribolium_castaneum/Info/Index, AMS_PRJEB1171_v1 [https://metazoa.ensembl.org/Adineta_vaga/Info/Index], GRCh37.p13 https://grch37.ensembl.org/Homo_sapiens/Info/Index, ASM20922v1 https://metazoa.ensembl.org/Nematostella_vectensis/Info/Index, Aqu1 https://metazoa.ensembl.org/Amphimedon_queenslandica/Info/Index, MneLei_Aug2011 http://metazoa.ensembl.org/Mnemiopsis_leidyi/Info/Index) or GIGA (PRJNA421033 https://www.ebi.ac.uk/ena/browser/view/PRJNA421033). Human Hi-C data was downloaded from NCBI (SRR1658570, HIC001). Processed files and tables needed to re-create the figures are accessible via a bitbucket repository: https://bitbucket.org/hannahschm/ceph_regulation_microsynteny/ . Source data are provided with this paper.

## Code availability

All bioinformatic protocols will be made available under https://bitbucket.org/hannahschm/ceph_regulation_microsynteny/ with detailed settings for each program and example scripts. The C++ script for 3D structures of individual chromosomes can be accessed upon request to T. Clarence (tereza.clarence@crick.ac.uk).

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

## Acknowledgements

H.S., O.P.H., E.R., and O.S. were supported by the Austrian Science Fund (FWF) grant P30686-B29. O.S. was supported by Whitman Center Early Career Fellowship (Frank R. Lillie Quasi-Endowment Fund, L. & A. Colwin Summer Research Fellowship, Bell Research Award in Tissue Engineering). H.S. was supported by the short-term grant abroad (KWA) of the University of Vienna. H.S. and O.S. were supported by the University of Chicago/ Vienna Strategic Partnership Programme Mobility Grant. A.K. was supported by the JSPS Postdoctoral Fellowship for Overseas Researchers program from Japan. C.B.A. was supported by the Hibbitt Early Career Fellowship. Eggs and paralarvae of *E. scolopes* were generated in part by support from the NASA Space Biology 80NSSC18K1465 awarded to J.S.F. S.V.N. was supported by the National Science Foundation IOS-1557914. This work was supported by the Francis Crick Institute, which receives its core funding from Cancer Research UK (FC0001003), the UK Medical Research Council (FC001003), and the Wellcome Trust (FC001003). Authors wish to thank Vienna Zoo (Tiergarten Schönbrunn), in particular, Roland Halbauer and the aquaristics team for animal husbandry, as well as the MBL Cephalopod Program, their team, Emily Garcia, and the MBL Central Microscopy Facility (MBL, Woods Hole). Authors thank the Department of Neuroscience and Developmental Biology at the University of Vienna, especially Andreas Denner. Computation was done using the Life Sciences Cluster at the University of Vienna. Sectioning was done at the Core Facility CIUS (University of Vienna). Authors wish to thank Daniel Rokhsar and Clifton Ragsdale for guidance and advice.

## Author contributions

H.S. performed comparative genomic analysis with the support of O.S., A.K. performed Hi-C experiments, H.S. and A.K. performed ATAC-seq data collection. H.S. and C.A. performed in situ hybridization. T.G., X.F., and P.B. performed 3D reconstruction and analysis. H.S. and O.P.H. performed ATAC-seq quantification. H.S. and B.Z. performed co-expression and syntenic randomization analysis, E.A.R., H.S., and C.A. performed RNA-seq extraction. J.F. and S.N. supplied animals. A.W. and Vienna Zoo housed animals. H.S., C.A., E.T., and O.S. designed the research. H.S. and O.S. wrote the manuscript with input from all other authors.

## Competing interests

The authors declare no competing interests.
