## [Peer Review File · Nature Communications]

REVIEWER COMMENTS

Reviewer #1 (Remarks to the Author):

The manuscript by Schmidbaur et al presents a comparative genomic analysis of gene microsynteny conserved at various evolutionary depths (across coleoid cephalopods, across molluscs, and across metazoa), and the relationships of microsynteny detected at different depths with genomic features (chromatin conformation from Hi-C, size of microsyntenic regions, size of introns and intergenic spaces within them) and the features of the genes they contain (GO overrepresented terms, expression patterns and their correlation).

The main conclusions are

- that coleoid cephalopod-specific genomic reorganisation and the resulting microsynteny are distinct from Metazoan and mollusc-wide ones
- that the explicit purpose of cephalopod genome reorganisation was to facilitate the cephalopod-specific nervous system innovations, namely the emergence of a large nervous system and complex behaviours in cephalopods

I don't think either of the conclusions is sufficiently supported by the results presented in the manuscript.

General remarks about the interpretation of the data:

- How can you be sure that these are cephalopod innovations rather than genomic regions with higher rearrangement rate relative to metazoan-level microsynteny?
- At an evolutionary distance/sequence divergence comparable to that of the analysed coleoid cephalopod genomes, other metazoan lineages have quite a lot of leftover syntenic regions that may be simply random - I.e. not enough time has passed since the last common ancestor for a more thorough shuffling.
- Indeed, despite the claims of tissue specificity of genes involved in cephalopod-only microsyntenic regions, gene ontology analysis (Extended data figure 3) comes up overwhelmingly with housekeeping terms. While I don't know what to think about their in-situ, the majority of genes in Figure 5d are also housekeeping (phenylalanine-tRNA ligase is a textbook example of a housekeeping protein). This makes one suspect that the normalisation of gene expression levels across different tissues might have been thrown off by different relative amounts of tissue-specific transcripts across different terminally differentiated cell types (e.g. in human expression tissues like whole blood and testis are notorious for that problem when compared against most other cell types), which may lead to systematically different baseline for housekeeping genes. The heatmap in Figure 5d is in agreement with this - the changes of levels of the five genes across the five tissues seem to be highly correlated. If you checked other housekeeping genes, I suspect most will be correlated with them, too. This would also provide a simple explanation of why genes in those microsyntenic regions are more highly expressed than those in the more ancient ones.
- Lines 167-179: In this part, the authors focus on two microsyntenic clusters (containing both cephalopod-specific and metazoan microsyntenic blocks) that showed preferential expression in nervous tissues. The authors then demonstrate that putative cis regulatory elements associated with these 2 clusters are enriched for transcription factor binding

motifs involved in nervous system differentiation. However, since these two clusters are mixed, the enrichment of tissue-specific regulatory elements can not be directly accounted to one specific class of microsynteny. This analysis would be clearer if the authors separated novel cephalopod-specific from ancient metazoan synteny and tested if any of the two classes show a higher density of tissue-specific regulatory elements.

Novelty and relation to previously published work

- Metazoan and mollusc-level microsynteny has telltale features of genomic regions previously described as genomic regulatory blocks (GRBs). This paper reads like the authors' knowledge of the concept and the associated literature is incomplete/fragmentary, and as a consequence reports as novel observations that have been made before.

- The authors mention "bystander model" and incorrectly cite Irimia et al (2012) as its source. The model in question is actually the GRB model, first suggested in Kikuta et al, *Genome Res* 2007, which includes the concept of "bystander gene" - a gene in (micro)synteny with a target gene of long-range enhancer-promoter interaction because its introns or flanking regions contain enhancers that control the nearby target gene. The target genes, not the bystander genes, are responsible for the overrepresentation of developmental and neuronal GO terms in such regions of microsynteny - both in vertebrates and e.g. in *Drosophila* (Engstrom et al *Genome Res* 2007). If the same holds in metazoan and mollusc-wide microsyntenic regions detected in cephalopods, it is highly likely the same underlying phenomenon.

- The authors cite the aforementioned Engstrom et al paper, which validated the GRB models in syntenic blocks preserved across the species of the *Drosophila* species, and show that microsynteny in those regions extends to much more distant mosquito genomes. The authors do not seem to have made the connection between the model described in that paper (and its consequences) with what they observe in cephalopods - although both the same phenomenon and the same functional classes of genes are involved.

- Lines 143-153: One of the direct consequences of the GRB model and the distinction between target and bystander genes is that genes within microsyntenic blocks do not necessarily have to be co-regulated to stay in cis. In this case, using average expression to study gene expression patterns associated with these blocks is probably not the best idea. Instead, the authors should consider parsing genes within microsyntenic blocks into groups with different tissue-specificity levels and analysing their expression patterns separately.

- Genomic regulatory blocks (GRBs) in other Metazoa have one defining feature whose existence should be easy to check in their hypothetical cephalopod equivalents: they contain clusters of highly conserved (ultraconserved) non-coding elements. The extent of the clusters of those elements predicts the extent of region that contains long-range enhancers, and also coincides with the span of TADs (Harmston et al. *Nat Comms* 2017). An analysis of noncoding elements highly conserved across coleoid cephalopods, and possibly also across molluscs, would be essential to establish the equivalence of cephalopod microsynteny blocks with the GRBs of other metazoan lineages. (I would not expect a significant number of noncoding elements conserved between cephalopods and more distantly related metazoa, just like there are practically none between e.g. vertebrates and insects). Comparing the identified clusters of conserved non-coding elements to the extent of cephalopod TADs would be a very important step towards establishing these genomic structures as ubiquitous across Metazoa.

- A comparison between two more closely related genomes (*O. bimaculoides* and *C. minor*)

would likely reveal a full range of GRB spans from the CNE content. It would also indicate whether cephalopod-specific microsyntenies also have increased density of CNEs (which would suggest that they are higher-turnover instances of the same process as the mollusc- and metazoan-wide ones), or if the CNEs are mostly absent, which might mean either a different mechanism for the maintenance of synteny or indeed no mechanism but just incomplete shuffling due to evolutionary proximity.

- Smaller intergenic spaces are also a possible indication that the colloid cephalopod-specific microsyntenies are to a significant extent random remnants of ancestral genome arrangement in regions not otherwise reliant in long-range promoter-enhancer interactions that span unrelated genes.

On the paper in general:

- The paper is not well written. It is surprisingly short for Nature Communications, like it was originally prepared for another journal with much tighter space constraints. In this case, brevity is not helping. Paragraphs do not connect well to each other, and each reads like a summary of a longer and likely more understandable text. An expansion of the text to provide more detail and context is essential. Good section titles would help, too.

- While it is good to learn what *E. scolopes* looks like, it is a bit unusual to have two opening figure panels (Figure 1a and 1b) serving no other purpose, especially since all the relevant bits from the Figure 1b are repeated in Figure 4e.

Minor points:

- There is a minor grammar mistake in the Abstract: "evolutionary unique gene clusters"" should be either "evolutionarily unique gene clusters"" , or "unique evolutionary gene clusters".

Reviewer #2 (Remarks to the Author):

The manuscript "Emergence of novel cephalopod gene regulation and expression through large-scale genome reorganization", by Schmidbaur and co-authors, uncovers genome-wide microsyntenic changes unique to cephalopod genomes and that are in some cases associated with the evolutionary innovations originated in this animal group. Employing a new chromosome-scale assembly of the squid *Euprymna scolopes* and a large set of metazoan genomes, the authors identify 505 unique cephalopod-specific microsyntenies, which are small syntenic blocks of a handful of genes with short intergenic regions. These cephalopod microsyntenies are scattered throughout the genome, suggesting a genome-wide reorganisation, and involve genes related to a variety of molecular functions and biological processes. To further characterise these microsyntenies, the authors go on to predict topological association domains (TADs), showing that microsyntenies tend to locate towards the centre of TADs, suggesting that there could be a pressure to maintain gene regulatory properties of these genes. However, the genes involved in cephalopod microsyntenic blocks did not show significant co-expression profiles. Instead, the authors analysed the averaged expression of a syntenic region (both cephalopod-specific and ancestral metazoan) and found 8 major groups of syntenic blocks defined by the expression dynamics in different cephalopod tissues, with modules 2 and 4 apparently dominated by cephalopod-specific syntenic blocks and largely expressed in neural tissues. Employing ATAC-seq data, the authors analysed the DNA motifs enriched in open chromatin regions of

cephalopod-specific microsynteny and the location of those putative regulatory regions, which appear to support the bystander model of gene regulation. Finally, the authors analyse one of the cephalopod-specific microsynteny clusters in greater detail, showing gene expression of the genes involved in neural tissue derivatives. Overall, the manuscript is a novel and important advance in our understanding of invertebrate and cephalopod genomes and opens an appealing avenue for exploring the evolutionary origins of some of the most remarkable morphological novelties in animals.

The data and analyses are of high quality and they are nicely presented. However, I think the text itself would benefit from a better structure/narrative. The questions/hypotheses that justify the data described in each paragraph are usually not clearly stated and most paragraph endings lack a summary/conclusion sentence (or these are presented in an isolated short paragraph). As such, some aspects of the manuscript come unjustified (e.g. why looking at 3D chromosomal modelling? How stable are chromosomal arrangements and how consistent these are between cell-types? could this modelling be influenced by the fact that the HiC data comes from a mixed population of cells?)

A central conclusion of the paper is that the large number of cephalopod-specific microsynteny clusters is related to the emergence of cephalopod innovations. However, there are also a large number of Octopoda-specific microsynteny clusters (301, Extended Data Fig 2a), which obviously appear after the origin of cephalopod-wide innovations. How do the authors interpret this finding? Could it be that the number of shared microsynteny clusters depends on the rate of evolutionary divergence between the lineages and/or time of divergence? It would be informative if all novel microsynteny clusters for each tree node are depicted in Extended Data Figure 2a, and if approximate divergence times are indicated (as in Fig 1c). Regarding the topology of the tree, which phylogenetic study was followed? It is strange to see *Adineta vaga* and *Schistosoma mansoni* as sister taxa.

The authors find no clear enrichment/association of individual genes of a cephalopod-specific microsyntenic block with a neural tissue/cephalopod innovation (lines 152-153). A putative association emerges when the expression of all genes in a microsyntenic block is averaged (Supp Note 7 line 432). Would similar associations emerge if expression of random syntenies is similarly averaged? I have the impression that by doing so, the authors are removing the intrinsic variability found between genes of a microsyntenic block, thus allowing patterns to emerge. E.g. it could be that the authors find an association of an averaged expression of a microsyntenic block to the brain, but that this association is mostly driven by just one gene that is very highly expressed in the brain, while the other genes in the block being expressed in other tissues but at a lower level. Is it possible to measure an index of gene expression variability among genes of a syntenic block, which could help to focus detail downstream analyses on only those clusters with more consistent expression among genes?

Following the previous point, what is the proportion of MACIs with respect to the 505 cephalopod-specific microsynteny clusters? The authors refer to clusters 2 and 4 as enriched (dominated, line 161) for cephalopod-specific microsynteny clusters, but that seems unclear from Fig. 3d. Can the author provide statistical evaluation that those clusters are indeed enriched in cephalopod-specific microsynteny clusters? Just from the colouring, it actually appears that it is other clusters (e.g. 1, 8 or 7) that contain more cephalopod-specific microsynteny clusters. Similarly, is there any particular reason to choose that MACI for further analyses? Without a better explanation, it seems cherry-picked, especially when genes show consistent expression in nervous system derivatives, but usually, and as the authors state, genes involved in cephalopod-specific microsynteny clusters are not co-expressed (see point above). All things considered, I think the general conclusion that cephalopod-specific microsynteny clusters and MACIs were crucial contributors to the evolution of cephalopod innovations might need

to be toned down.

Other comments:

- **microsyntenies associated with cephalopod innovations (MACIs).** It is odd that the authors refer to them in the abstract but this term comes only at the very end of the manuscript.
- **Lines 41-45:** the sentence appears to assume that cephalopod genomes have altered their genome organisation and regulation, but that is only stated a couple of sentences below, and thus the sentence reads strange.
- **Line 49:** reference needed for the breakage and addition of novel microsyntenies in cephalopods.
- **Line 54-55:** Figure 1c (a cladogram) is reference in the context of microsyntenic complement. Might it be more adequate to reference it together with Figure 1a,b?
- **Extended Data Figure 2a** states 500 cephalopod-specific microsyntenies, but the text states 505.
- **Extended Data Figure 2b, c** do not clearly show the differences/similarities between microsyntenic types. Probably another type of visualisation (e.g. violin plot plus a box plot indicating the median or so) will aid.
- **Do novel microsyntenies involve cephalopod-specific genes?**
- **Lines 93-96:** Where is that data (71% of *M. yessoensis* single copy orthologs are located in different chromosomes) shown? There is a type with *M. yessoensis* in that sentence too.
- **Lines 103-106:** these two sentences are somehow redundant.
- **Extended Data 4a-c:** bootstrap values are not shown (at least those of high value). How is the CTCF tree rooted?
- **Line 121:** "Since novel microsyntenies are transcriptionally active". Where is that shown?
- There is some formatting issue with Supplementary Material, as some symbols (e.g. -) are not recognised.

Chema Martin

Reviewer #3 (Remarks to the Author):

I read the manuscript "Emergence of novel cephalopod gene regulation and expression through large-scale genome reorganization" by Hannah Schmidbaur and co-authors. The study address a very interesting scientific finding, i.e. the emergence within cephalopod molluscs of unique genomic, topological, and regulatory organization of cephalopod genomes. In particular, the Authors confirm the existence of hundreds of microsyntenies, identifying as corresponding to "topological compartments with a distinct regulatory structure", contributing to complex expression patterns associated with cephalopod innovations that appeared enriched in cephalopod nervous system expression. I agree with the Authors conclusion that these innovations, associated with microsyntenies, appear to be instrumental to the evolution of cephalopod nervous system complexity. To support their findings Authors approached a compelling series of analyses, aimed at describing topological and regulatory genome organization in one cephalopod species, namely *Euprymna scolopes*. Authors also attempted a comparison with other selected metazoan species, to further support their findings.

I found the manuscript easy to read in the great majority of cases. Sometimes it is affected by a lack of proper punctuation in several sentences that make it quite difficult to read. There are also some typo, thus I suggest that the Authors will read with attention the whole text - including Supplementary Notes - and revise their English in order to further improve the overall quality of the manuscript.

I have few notes and comments for the Authors to consider in a revision.

1. The results of in situ hybridization experiments (FISH and ISH) lack of a control image to show. I suggest that the authors extend the data shown in Fig. 4e by adding some magnification and/or additional images in a more comprehensive figure to be included as Supplementary Material. As shown figures are not very informative.

1.1. Scale bar is added in Fig. 4e, but there is no indication in the figure legend.

1.2 Authors should indicate the stage (they refer to late E. scolopes embryos).

1.3 A control section/image for each of the genes considered is suggested to be added in the additional supplementary figure requested.

1.4. Supplementary Figure Extended Data Fig. 8 (ISH): the identification of areas within the brain is not possible through the image; Authors need to identify better areas, even though this is an embryonic brain, considering that Authors are using late embryos (as reported) it is possible to identify specific areas/masses within the so-called "brain". This will provide further relevance to the finding, despite the limitation in the number of genes selected.

1.5 This reviewer finds difficult to understand the rationale for using the five genes

1.6 in the Supplementary Figure (Extended Data Fig. 8) legend abbreviations refer to structures that are not depicted or where no signal is observed.

Please amend all Figures, Legends accurately

1.7 Overall the data shown after the included in situ hybridization experiments require a more accurate description and a more accurate representation, including representative additional sections, and more detailed magnifications to better show specificity of signal observed; this also requires control sections to be shown.

2. Use of Live Animals and sample collection.

2.1 Despite Authors obtained live animals/embryos from labs in the USA, it is highly recommended to refer to ARRIVE checklist (<https://arriveguidelines.org/arrive-guidelines>) that Nature Journals advice to follow.

Many details are missing in this version of the manuscript.

It is strongly advised to modify the relevant sections.

2.2 Experiments and live animals obtained not in EU are not following Directive 2010/63/EU. Nevertheless, following ARRIVE checklist and in general considering that Cephalopods are included in the Directive 2010/63/EU a statement about animal welfare and ethical compliance of animals care should be provided.

3. Minor:

3.1 Supplementary Notes #1. Authors should refer to Collins, A.J., and Nyholm, S.V. (2010). Obtaining Hemocytes from the Hawaiian Bobtail Squid *Euprymna scolopes* and Observing their Adherence to Symbiotic and Non-Symbiotic Bacteria. *J Vis Exp* (36), e1714. when referring to "anaesthesia" applied to these animals and to CSHL Protocols available for the species.

The reference to Fiorito et al., 2015 (Guidelines for animal care) is a general one; it maybe utilized here only to refer to the slow increase of solutes to reach the final concentration in the anaesthetic solution.

3.2 "p" are indicated in several instances; a sentence or a reference to the statistics adopted to obtain that p value will be helpful for the Reader (somewhere in the text, summarize statistics; it is diluted in the text somewhere and is not clear enough). Only in one figure legend I can trace reference to statistical tests adopted

3.3 Lines 137-141 Main text; the sentence reads not very clear.

3.4 I am unable to trace BioProject data, Authors declare to being deposited. Please amend the sentence or correct UID

3.5 Supplementary Notes

- a. Please use headings of tables repeating in the following pages to facilitate readability**
- b. Table 2: sort by taxonomy; add taxa (these does not necessarily match with the logic of inclusion of Table 3)**
- c. Table 3: includes/mention Rotifera in the Lophotrochozoa, please correct**
- d. better justification for not using hemocytes data is required. The low abundance of transcripts is novel/surprising to this reviewer, and requires justification**
- e. Table 6 indicate F/R**
- f. beta-tubulin relative abundance in tissues considered should be mentioned somewhere in supplementary material at the late embryo stage considered. It is not always homogeneous in tissues to be reported as "control"**

Minor

Reviewer #4 (Remarks to the Author):

As an evolutionary biologist, I reviewed this manuscript in the context of broad patterns of evolution and implications for cephalopod genomics. Overall, I think this is a very well written paper with rigorous analyses to support the author's major hypotheses. I spent more time re-evaluating the paper to determine if I could provide any additional useful critique - I could not.

My only significant comment relates to the introduction:

Line 47-49: This paragraph is oddly worded. It would be helpful to the reader to have it re-phrased as it left me unclear about the current status of this type of knowledge in cephalopods. E.g.

- Has evidence for genome breakage and addition of novel microsyntenies in cephalopods been published? This statement currently reads as this information is pre-existing. I cannot recall if this work was addressed in the two cephalopod genome papers cited?**
- Same comment for the concluding sentence. Cite any previous work where this type of question was addressed. Have the extent and impact of arrangements not been investigated, or did data/etc. limitations make asking such questions impossible? This seems an important point to emphasize. It would better frame the significant contribution this work provides to the field of cephalopod genomics.**

REVIEWER COMMENTS

We thank the editor and the four referees for their detailed comments, which helped us improve our manuscript. Overall, we have made substantial edits and expanded the manuscript to increase its readability and more clearly discuss our findings in the context of known genomic regulatory blocks. We have added several analyses, including new *in situ* sectioning, a new Figure 5, conserved non-coding element analyses and more in-depth co-expression analyses. Below we respond to the specific referees' comments.

Reviewer #1 (Remarks to the Author):

The manuscript by Schmidbaur et al presents a comparative genomic analysis of gene microsynteny conserved at various evolutionary depths (across coleoid cephalopods, across molluscs, and across metazoa), and the relationships of microsynteny detected at different depths with genomic features (chromatin conformation from Hi-C, size of microsyntenic regions, size of introns and intergenic spaces within them) and the features of the genes they contain (GO overrepresented terms, expression patterns and their correlation).

The main conclusions are

- that coleoid cephalopod-specific genomic reorganisation and the resulting microsynteny are distinct from Metazoan and mollusc-wide ones
- that the explicit purpose of cephalopod genome reorganisation was to facilitate the cephalopod-specific nervous system innovations, namely the emergence of a large nervous system and complex behaviours in cephalopods

I don't think either of the conclusions is sufficiently supported by the results presented in the manuscript.

General remarks about the interpretation of the data:

- How can you be sure that these are cephalopod innovations rather than genomic regions with higher rearrangement rate relative to metazoan-level microsynteny?

Response: We infer cephalopod novel microsynteny using phylogenetic sampling of many species, which show that those syntenies are absent in other animal genomes, except for cephalopods. This by itself is one side of the synapomorphic characteristic of these regions. Secondly, we now provide randomization data (Extended Data Figure 2) which shows that false-positive detection rate of microsynteny is minimal with our inference methods and cutoffs used. We rephrased the text to make it more clear, specifically:

“Briefly, we define microsyntenic blocks as at least three or more co-occurring orthologous genes with up to five intervening genes, with no constraints on their collinearity. This definition of microsynteny yields the fewest false-positive blocks (compared to just pairs of genes) while

providing enough flexibility to detect syntenic regions that underwent local rearrangement and expansion. We recover 505 microsynteny unique to cephalopods, representing blocks of genes only found in close proximity to each other in E. scolopes and at least one octopus species. For the same species sampling and same microsynteny detection parameters only 2 blocks would have been expected by chance (median from 3 rounds of randomization, as described in ⁶). Five out of these 505 blocks were paralogous.”

Additionally, we have conducted simulations of microsynteny using random sampling of regions to mimic the distribution of detected microsynteny and find that those random regions have different expression and regulatory properties. While we do not know recombination rates within cephalopods, such distinct properties, together with the observed preservation of at least a subset of ancient animal microsynteny, suggest different selective pressures.

- At an evolutionary distance/sequence divergence comparable to that of the analysed coleoid cephalopod genomes, other metazoan lineages have quite a lot of leftover synteny that may be simply random - I.e. not enough time has passed since the last common ancestor for a more thorough shuffling.

Response: In our analyses, coleoid cephalopod obtained most (500) of novel microsynteny, whereas the background molluscan gain rate (e.g., between *Aplysia* and *Lottia*) is only 43. This large number is also not expected by chance in our microsynteny detection. The reviewer is right that when we look at gene pairs, the false-positive retention rates are much higher (therefore we use cutoffs of at least 3 genes to define microsynteny). In that case, we did not observe any microsynteny.

- Indeed, despite the claims of tissue specificity of genes involved in cephalopod-only microsynteny regions, gene ontology analysis (Extended data figure 3) comes up overwhelmingly with housekeeping terms. While I don't know what to think about their in-situ, the majority of genes in Figure 5d are also housekeeping (phenylalanine-tRNA ligase is a textbook example of a housekeeping protein). This makes one suspect that the normalisation of gene expression levels across different tissues might have been thrown off by different relative amounts of tissue-specific transcripts across different terminally differentiated cell types (e.g. in human expression tissues like whole blood and testis are notorious for that problem when compared against most other cell types), which may lead to systematically different baseline for housekeeping genes. The heatmap in Figure 5d is in agreement with this - the changes of levels of the five genes across the five tissues seem to be highly correlated. If you checked other housekeeping genes, I suspect most will be correlated with them, too. This would also provide a simple explanation of why genes in those microsyntenic regions are more highly expressed than those in the more ancient ones.

Response: We are not claiming that genes in novel microsynteny have higher expression than other genes. Indeed, the relative expression puts them into different expression categories (Figure 4). Moreover we find similar patterns of expression module correspondence in octopus

(EDF7). We do provide co-expression quantification between novel cephalopod, metazoan, and random blocks in Figure 3, showing distinct co-expression (of normalized block expression). While the reviewer is right that often there is a dominant gene per microsyntenic block, averaged relative expression shows consistency of tissue contribution among expression modules.

Additionally, the overall normalization of our matrices is likely not the reason for this pattern as we observe very distinct expression modules which are also confirmed by the expression analysis of their orthologs in octopus.

We have extended this section to include a more balanced discussion of the absolute expression levels:

“We next wanted to investigate whether these expression patterns are biased due to a single highly expressed gene per syntenic block. The vast majority of microsyntenic blocks (76%, Extended Data Figure 8b) had one gene that contributed to more than 50% of the cumulative expression. We then calculated relative expression levels across tissues per gene and averaged it for each block in the expression modules, showing that overall expression module identities are retained (Extended Data Figure 8a). In general, expression variance correlates with absolute expression of genes (Extended Data Figure 8c). In some metazoan syntenies, however, especially in the tissues defining a module, the variance was low, indicating higher co-expression constraints. (Extended Data Figure 8c).”

- Lines 167-179: In this part, the authors focus on two microsyntenic clusters (containing both cephalopod-specific and metazoan microsyntenic blocks) that showed preferential expression in nervous tissues. The authors then demonstrate that putative cis regulatory elements associated with these 2 clusters are enriched for transcription factor binding motifs involved in nervous system differentiation. However, since these two clusters are mixed, the enrichment of tissue-specific regulatory elements can not be directly accounted to one specific class of microsynteny. This analysis would be clearer if the authors separated novel cephalopod-specific from ancient metazoan syntenies and tested if any of the two classes show a higher density of tissue-specific regulatory elements.

Response: We apologize for the confusion. We discuss cephalopod microsynteny and expression and separate it from the neighboring metazoan microsyntenic cluster. We have made changes to the text to make it clearer, specifically:

“The cluster was localized towards the center of the predicted TAD (Figure 4b), close to another novel microsyntenic unit (from the same expression module). Together, these two units form a compartment of high Hi-C interaction density, separate from the closest metazoan microsyntenic compartment and its associated ATAC-peaks (Figure 4b).”

In respect to motif enrichment analyses, these were done on all cephalopod and metazoan microsyntenic clusters, respectively. This seems to be in accord with the request and we apologize again for failing to make our statements clear. Specifically, we modified the text to:

“Predicted peaks associated with each expression cluster were then further analysed for known transcription factor motif enrichment, separately for cephalopod and ancient microsynteny (Methods, Extended Data Table 1).”

Novelty and relation to previously published work

- Metazoan and mollusc-level microsynteny have telltale features of genomic regions previously described as genomic regulatory blocks (GRBs). This paper reads like the authors' knowledge of the concept and the associated literature is incomplete/fragmentary, and as a consequence reports as novel observations that have been made before.

Response: We thank the reviewer for this comment. We use the term microsynteny in our manuscript primarily, instead of GRB, to highlight the fact that they have been inferred solely through comparative genomics (and thus not all may be functional GRBs). We agree that this requires more careful elaboration on our part and we have substantially extended the appropriate sections as well as the introduction. In fact, we are glad that the reviewer finds that our microsynteny and their regulatory and expression assessment to be reminiscent of GRBs. They thus could indeed be regulatory blocks worth of further investigation and regulatory dissection. We would still like to remain on the safe side in calling them primarily microsynteny, as microsyntenic detection is indeed affected by genetic divergence, length of the microsyntenic block etc. (see also other comments). In general, we did not claim to discover GRBs (or microsynteny), but we do describe them in cephalopods and link their emergence to the unique genome reshuffling and genome regulatory landscape, which would be a novel finding at least for this clade of animals.

- The authors mention "bystander model" and incorrectly cite Irimia et al (2012) as its source. The model in question is actually the GRB model, first suggested in Kikuta et al, Genome Res 2007, which includes the concept of "bystander gene" - a gene in (micro)synteny with a target gene of long-range enhancer-promoter interaction because its introns or flanking regions contain enhancers that control the nearby target gene. The target genes, not the bystander genes, are responsible for the overrepresentation of developmental and neuronal GO terms in such regions of microsynteny - both in vertebrates and e.g. in *Drosophila* (Engstrom et al Genome Res 2007). If the same holds in metazoan and mollusc-wide microsyntenic regions detected in cephalopods, it is highly likely the same underlying phenomenon.

Response: Thank you, we fixed the proper citation. Indeed, our manuscript, while analyzing microsyntenic clusters, suggests their functional role and their regulatory properties. Our analysis thus helps reveal putative cephalopod GRBs and their type. We have clarified this in the text. We find that while intron size remains similar in ancient and cephalopod microsynteny, cephalopod microsynteny tend to have much smaller intergenic distances. Our finding of ATAC peak enrichment in the intronic sequences (and indeed of our main example from Figures 4 and 5 - the peak is located in the "metabolic" tRNA gene) is thus also

reminiscent of the GRB model. We made this elaboration throughout the text, and specifically in the new “Evolutionary scenarios for functional microsynteny emergence in cephalopods” section:

“This result indicates that, unlike more conserved microsynteny, novel cephalopod microsynteny is similar to the GRB scenario, in particular the bystander model, in which putative enhancers are found within closely located (bystander) genes inside the same microsyntenic cluster and may potentially lack distal regulatory domains.”

- The authors cite the aforementioned Engstrom et al paper, which validated the GRB models in syntenic blocks preserved across the species of the *Drosophila* species, and show that microsynteny in those regions extend to much more distant mosquito genomes. The authors do not seem to have made the connection between the model described in that paper (and its consequences) with what they observe in cephalopods - although both the same phenomenon and the same functional classes of genes are involved.

Response: Thank you. See comments above, indeed it was our intention to make such a connection and we have extended the text in an attempt to make it more correct and understandable.

- Lines 143-153: One of the direct consequences of the GRB model and the distinction between target and bystander genes is that genes within microsyntenic blocks do not necessarily have to be co-regulated to stay in cis. In this case, using average expression to study gene expression patterns associated with these blocks is probably not the best idea. Instead, the authors should consider parsing genes within microsyntenic blocks into groups with different tissue-specificity levels and analysing their expression patterns separately.

Response: Our analysis indeed finds significantly less co-expression in novel cephalopod microsynteny, indicative of the GRB model. Due to the nature of randomization it would not be possible for us to provide for a proper control for co-expression within the tissue-specific levels. Additionally, tissue sampling will affect this calculation. We do however show in the heatmap (Figure 4) different modules of varying tissue specificity and we show their retention in octopus tissues as well (EDF7), which indicates evolutionary conservation. We have also now added a new paragraph where we discuss the dissimilarities in the absolute expression of genes within each microsyntenic block (as well as a new EDF8):

“We next wanted to investigate whether these expression patterns are biased due to a single highly expressed gene per syntenic block. The vast majority of microsyntenic blocks (76%, Extended Data Figure 8b) had one gene that contributed to more than 50% of the cumulative expression. We then calculated relative expression levels across tissues per gene and averaged it for each block in the expression modules, showing that overall expression module identities are retained (Extended Data Figure 8a). In general, expression variance correlates with absolute expression of genes (Extended Data Figure 8c). In some metazoan syntenies, however, especially in the tissues defining a module, the variance was low, indicating higher co-expression constraints. (Extended Data Figure 8c).”

- Genomic regulatory blocks (GRBs) in other Metazoa have one defining features whose existence should be easy to check in their hypothetical cephalopod equivalents: they contain clusters of highly conserved (ultraconserved) non-coding elements. The extent of the clusters of those elements predicts the extent of region that contains long-range enhancers, and also coincides with the span of TADs (Harmston et al. Nat Comms 2017). An analysis of noncoding elements highly conserved across coleoid cephalopods, and possibly also across molluscs, would be essential to establish the equivalence of cephalopod microsynteny blocks with the GRBs of other metazoan lineages. (I would not expect a significant number of noncoding elements conserved between cephalopods are more distantly related metazoa, just like there are practically none between e.g. vertebrates and insects). Comparing the identified clusters of conserved non-coding elements to the extent of cephalopod TADs would be a very important step towards establishing these genomic structures as ubiquitous across Metazoa.

Response: Thank you for this valuable comment. We have now included conserved non-coding element analysis (EDF9) and indeed we can show that novel cephalopod microsynteny have large intronic ATAC retention (Fig 3). We do, however, find surprisingly little overlap between CNEs and ATAC data which may indicate many species-specific regulatory regions. We report our findings in a new paragraph:

“To complement our ATAC data, we conducted whole genome alignments between available cephalopod genomes to identify potential conserved non-coding elements (CNEs, Methods, Extended Data Figure 9, Supplementary Note 8). Due to evolutionary distance between squids and octopus, our approach yielded only 662 coleoid cephalopod CNE candidates, of which 77 were associated with novel cephalopod microsynteny, 41 with metazoan syntenies, and 209 were located outside any synteny. Only six had overlap with ATAC peaks (Extended Data Figure 9b). For CNEs shared among squid genomes we found 47013 putative CNEs, of which 2987 were associated with novel cephalopod microsynteny, 3691 with metazoan synteny, and 9920 were located around genes outside any synteny (Extended Data Figure 9c). Similarly, very few overlapped with ATAC peaks (61). Therefore the regulatory role of these regions remains unclear (see below).”

Despite little overlap, we do however find statistical enrichment of CNEs towards more distal regions around ancient animal microsynteny:

“The location of ATAC-seq peaks within the microsyntenic clusters reveals that open chromatin regions associated with cephalopod-specific microsynteny were more often found in introns than peaks found in conserved, metazoan microsynteny or other non-syntenic genes (Figure 3e, Fisher’s exact test $p < 1e-5$). Interestingly, while sharing little overlap, CNE distribution, in particular of CNEs shared among squids, showed significant localisation towards distal regions in metazoan syntenies (Fisher’s exact test $p < 2.2e-16$) (Extended Data Figure 9).”

This result may be impacted by the fact that ATAC-seq was performed on developmental stages as well as often little genomic conservation which CNE detection relies on. We thus expect that availability of more closely related genomes will help find a consensus. Nevertheless, the higher conservation in distal regions of ancient microsynteny may indicate their role as master regulatory regions of those microsynteny (and thus a co-expression model of microsynteny).

- A comparison between two more closely related genomes (*O. bimaculoides* and *C. minor*) would likely reveal a full range of GRB spans from the CNE content. It would also indicate whether cephalopod-specific microsynteny also have increased density of CNEs (which would suggest that they are higher-turnover instances of the same process as the mollusc- and metazoan-wide ones), or if the CNEs are mostly absent, which might mean either a different mechanism for the maintenance of synteny or indeed no mechanism but just incomplete shuffling due to evolutionary proximity.

Response: We have included analysis of CNEs using several species, mainly focusing on squid lineages: *Architeuthis dux* and *E. scolopes*. Already in the squid lineage we find many more retained putative CNEs than between squid and octopus, however the false-positive rate of such detection is unclear and is likely to be even higher for the relatively closely related octopus species.

- Smaller intergenic spaces are also a possible indication that the colloid cephalopod-specific microsynteny are to a significant extent random remnants of ancestral genome arrangement in regions not otherwise reliant in long-range promoter-enhancer interactions that span unrelated genes.

Response: Thank you for this comment, this may be the case, but it is still worth mentioning that such small intergenic distances are very uncharacteristic of a very large genome, indicating some selective pressure and supporting the “functional” microsynteny, i.e., the GRB/bystander model. Almost any genome that has been studied has a strong correlation of intron and intergenic sizes with the total genome size. Coleoid cephalopod-specific microsynteny thus show a different pattern.

On the paper in general:

- The paper is not well written. It is surprisingly short for Nature Communications, like it was originally prepared for another journal with much tighter space constraints. In this case, brevity is not helping. Paragraphs do not connect well to each other, and each reads like a summary of a longer and likely more understandable text. An expansion of the text to provide more detail and context is essential. Good section titles would help, too.

Response: Thank you, we made extensive efforts to improve our writing, extending the text to conform with the journal expectations, introducing (more) subsections as well as new main and extended data figures.

- While it is good to learn what *E. scolopes* looks like, it is a bit unusual to have two opening figure panels (Figure 1a and 1b) serving no other purpose, especially since all the relevant bits from the Figure 1b are repeated in Figure 4e.

Response: We have modified Figure 1 and moved the panel into the new Figure 5 where it is used to navigate the expression patterns.

Minor points:

- There is a minor grammar mistake in the Abstract: "evolutionary unique gene clusters"" should be either "evolutionarily unique gene clusters"", or "unique evolutionary gene clusters".

Response: Thank you, fixed.

Reviewer #2 (Remarks to the Author):

The manuscript "Emergence of novel cephalopod gene regulation and expression through large-scale genome reorganization", by Schmidbaur and co-authors, uncovers genome-wide microsyntenic changes unique to cephalopod genomes and that are in some cases associated with the evolutionary innovations originated in this animal group. Employing a new chromosome-scale assembly of the squid *Euprymna scolopes* and a large set of metazoan genomes, the authors identify 505 unique cephalopod-specific microsyntenies, which are small syntenic blocks of a handful of genes with short intergenic regions. These cephalopod microsyntenies are scattered throughout the genome, suggesting a genome-wide reorganisation, and involve genes related to a variety of molecular functions and biological processes. To further characterise these microsyntenies, the authors go on to predict topological association domains (TADs), showing that microsyntenies tend to locate towards the centre of TADs, suggesting that there could be a pressure to maintain gene regulatory properties of these genes. However, the genes involved in cephalopod microsyntenic blocks did not show significant co-expression profiles. Instead, the authors analysed the averaged expression of a syntenic region (both cephalopod-specific and ancestral metazoan) and found 8 major groups of syntenic blocks defined by the expression dynamics in different cephalopod tissues, with modules 2 and 4 apparently dominated by cephalopod-specific syntenic blocks and largely expressed in neural tissues. Employing ATAC-seq data, the authors analysed the DNA motifs enriched in open chromatin regions of cephalopod-specific microsyntenies and the location of those putative regulatory regions, which appear to support the bystander model of gene regulation. Finally, the authors analyse one of the cephalopod-specific microsyntenies of cluster 2 in greater detail, showing gene expression of the genes involved in neural tissue derivatives. Overall, the manuscript is a novel and important advance in our understanding of invertebrate and cephalopod genomes and opens an appealing avenue for exploring the evolutionary origins of some of the most remarkable morphological novelties in animals.

Response: Thank you

The data and analyses are of high quality and they are nicely presented. However, I think the text itself would benefit from a better

structure/narrative. The questions/hypotheses that justify the data described in each paragraph are usually not clearly stated and most paragraphs

endings lack a summary/conclusion sentence (or these are presented in an isolated short paragraph). As such, some aspects of the manuscript come

unjustified (e.g. why looking at 3D chromosomal modelling? How stable are chromosomal arrangements and how consistent these are between cell-types?

could this modelling be influenced by the fact that the HiC data comes from a mixed population of cells?)

Response: Thank you, we have substantially reworked and extended the manuscript. We hope that our edits help increase its readability. We introduce the implementation of methods more carefully now. In respect to the specific comment on chromosomal modeling: while we cannot do a single-cell analysis of (whole) cephalopods (yet), 3D modelling allows us to understand the localization and interactions of evolutionarily novel regions. We have now elaborated why 3D location of synteny in respect to their global regulation is important. It is a shortcoming of the current multicell method that we cannot identify smaller cell-type specific organization/chromosomal folding, but our simulation method (EDF5) is consistent and it thus can be used to infer the global chromosomal localization of synteny.

As it is common for tissue development and cell differentiation, we believe we could expect movements of TAD boundaries or TAD merging (1, 2), therefore the resulting model will reflect those features. We do expect a certain level of chromatin landscape remodelling across tissues, to what extent this can impact the surface location of synteny remains to be explored. Also, if the remodelling occurred on scale within 150kb bins, this model would not be able to capture it unless we had finer Hi-C resolution.

1). Long HK, Prescott SL, Wysocka J. Ever-Changing Landscapes: Transcriptional Enhancers in Development and Evolution. *Cell*. 2016 Nov 17;167(5):1170-1187. doi:

10.1016/j.cell.2016.09.018. PMID: 27863239; PMCID: PMC5123704. 2). Hug CB, Vaquerizas JM. The Birth of the 3D Genome during Early Embryonic Development. *Trends Genet*. 2018 Dec;34(12):903-914. doi: 10.1016/j.tig.2018.09.002. Epub 2018 Oct 3. PMID: 30292539.

A central conclusion of the paper is that the large number of cephalopod-specific microsynteny is related to the emergence of cephalopod innovations. However, there are also a large number of Octopoda-specific microsynteny (301, Extended Data Fig 2a), which obviously appear after the origin of cephalopod-wide innovations. How do the authors interpret this finding? Could it be that the number of shared microsynteny depends on the rate of evolutionary divergence between the lineages and/or time of divergence? It would be informative if all novel microsynteny for each tree node are depicted in Extended Data Figure 2a, and if approximate divergence times are indicated (as in Fig 1c). Regarding the topology of the tree, which phylogenetic study was followed? It is strange to see *Adineta vaga* and *Schistosoma mansoni* as sister taxa.

Response: Thank you for this comment. We agree, the large number of octopus-specific microsynteny is indeed striking. While there certainly is a relationship between the number of microsynteny and evolutionary divergence, we see far fewer microsynteny in comparably diverged clades (e.g. chordates, ambulacrarians relative to coleoids). We have added the divergence time and number of novel microsynteny at many of the nodes in EDF2a. Notably, at a similar genetic distance, only (coleoid) cephalopods show such a higher gain of novel microsynteny. We have also added randomization analysis to reveal that, by chance, very few false-positive microsynteny are detected.

The authors find no clear enrichment/association of individual genes of a cephalopod-specific microsyntenic block with a neural tissue/cephalopod innovation (lines 152-153). A putative association emerges when the expression of all genes in a microsyntenic block is averaged (Supp Note 7 line 432). Would similar associations emerge if expression of random syntenies is similarly averaged? I have the impression that by doing so, the authors are removing the intrinsic variability found between genes of a microsyntenic block, thus allowing patterns to emerge. E.g. it could be that the authors find an association of an averaged expression of a microsyntenic block to the brain, but that this association is mostly driven by just one gene that is very highly expressed in the brain, while the other genes in the block being expressed in other tissues but a lower level. Is it possible to measure an index of gene expression variability among genes of a syntenic block, which could help to focus detail downstream analyses on only those cluster with more consistent expression among genes?

Response: We agree with the reviewer that averaged expression could mainly be driven by one gene when considering absolute expression counts per microsyntenic block. This is also the case in our data where usually one gene demonstrates higher expression than others. We have now quantified the average expression based on relative (not absolute) gene expression to give each gene in a microsyntenic block the same weight. The analysis shows that overall the tissue expression patterns of the expression clusters 1-8 from Fig 3 are retained. Additionally, as a measure of variability among genes within syntenic blocks we looked at the mean relative expression in each syntenic block vs. its standard deviation. In most cases, the higher the mean relative expression in a given tissue, the higher the standard deviation between samples, which is expected. This analysis shows that cephalopod syntenies sometimes deviate from the overall pattern in that they have genes with high expression but less deviation.

Following the previous point, what is the proportion of MACIs with respect to the 505 cephalopod-specific microsynteny? The authors refer to clusters 2 and 4 as enriched (dominated, line 161) for cephalopod-specific microsynteny, but that seems unclear from Fig. 3d. Can the author provide statistical evaluation that those clusters are indeed enriched in cephalopod-specific microsynteny? Just from the colouring, it actually appears

that it is other clusters (e.g. 1, 8 or 7) that contain more cephalopod-specific microsyntenies. Similarly, is there any particular reason to choose that MACI for further analyses? Without a better explanation, it seems cherry-picked, specially when genes show consistent expression in nervous system derivatives, but usually, and as the authors state, genes involved in cephalopod-specific microsyntenies are not co-expressed (see point above). All things considered, I think the general conclusion that cephalopod-specific microsyntenies and MACIs were crucial contributors to the evolution of cephalopod innovations might need to be toned down.

Response: We have added barplots showing relative ratios of cephalopod and metazoan syntenies in each of the expression modules. In terms of microsyntenies most contributing to nervous system expression (modules 2 and 4), we have in total 128 ceph and only 44 metazoan syntenies in expression module 2 and 114 and 18 syntenies in expression module 4, respectively. According to the overall expectation and Fisher's exact test, this is significant enrichment. To clarify that we have added the following sentence:

“Interestingly, clusters encompassing multiple nervous tissues like modules 2 and 4 were enriched in novel cephalopod microsyntenies (Fisher's exact test p-values ≤ 0.02 and $\leq 1e-07$, respectively).”

As to the choice of the microsyntenic block for the in situs: it was the longest and within the expression module most dominated by MACIs and also of relevance to the nervous system. We decided to investigate its expression to confirm its nervous tissue affiliation. Indeed, co-expression is defined through all tissues and in situ, while showing overlap in some tissues, indicates areas of unique expression.

Other comments:

- microsyntenies associated with cephalopod innovations (MACIs). It is odd that the authors refer to them in the abstract but this term comes only at the very end of the manuscript.

Response: Fixed

- Lines 41-45: the sentence appears to assume that cephalopod genomes have altered their genome organisation and regulation, but that is only stated a couple of sentences below, and thus the sentence reads strange.

Response: Thank you. We have rephrased the introduction

- Line 49: reference needed for the breakage and addition of novel microsyntenies in cephalopods.

Response: Thank you. We have added the reference

- Line 54-55: Figure 1c (a cladogram) is reference in the context of microsyntenic complement. Might it be more adequate to reference it together with Figure 1a,b?

Response: We have reworked this section and also note Extended Data Figure 2 tree:

“These microsyntenies have been conserved in coleoid cephalopods despite their long divergence time (Figure 1b, c)” and “(Figure 1c, Extended Data Figure 2a, Methods)”

- Extended Data Figure 2a states 500 cephalopod-specific microsyntenies, but the text states 505.

Response: There are 505, but 5 are paralogous; we have now stated this more clearly

- Extended Data Figure 2b, c do not clearly show the differences/similarities between microsyntenic types. Probably another type of visualisation

(e.g. violin plot plus a box plot indicating the median or so) will aid.

Response: Thank you, we have added this information.

- Do novel microsyntenies involve cephalopod-specific genes?

Response: We have added orphan gene analysis:

“In total, only 48 out of 2290 genes in these 505 blocks were identified as orphan genes with no homology outside of cephalopods, while all others have orthologs in other animals, suggesting that the origin of microsynteny was due to changes in gene locations rather than novel gene emergence.”

- Lines 93-96: Where is that data (71% of *M. yessoensis* single copy orthologs are located in different chromosomes) shown? There is a type with *M. yessoensis* in that sentence too.

Response: We have rephrased that sentence to make it clear where the counts come from.

- Lines 103-106: these two sentences are somehow redundant.

Response: We have rephrased.

- Extended Data 4a-c: bootstrap values are not shown (at least those of high value). How is the CTCF tree rooted?

Response: We have added bootstrap information to the figure. The tree is unrooted

- Line 121: "Since novel microsyntenies are transcriptionally active". Where is that shown?

Response: We have rephrased to:

“Since the novel microsyntenies are transcriptionally active (Figure 3, see below), their location on the chromosomal surface may be reflective of highly dynamic inter-chromosomal regulation, as well as being more accessible to transcription factors.”

- There is some formatting issue with Supplementary Material, as some symbols (e.g. -) are not recognised.

Response: Thank you, we fixed formatting issues in the supplement.

Chema Martin

Reviewer #3 (Remarks to the Author):

I read the manuscript "Emergence of novel cephalopod gene regulation and expression through large-scale genome reorganization" by Hannah Schmidbaur and co-authors. The study address a very interesting scientific finding, i.e. the emergence within cephalopod molluscs of unique genomic, topological, and regulatory organization of cephalopod genomes. In particular, the Authors confirm the existence of hundreds of microsyntenies, identifying as corresponding to "topological compartments with a distinct regulatory structure", contributing to complex expression patterns associated with cephalopod innovations that appeared enriched in cephalopod nervous system expression.

I agree with the Authors conclusion that these innovations, associated with microsyntenies, appear to be instrumental to the evolution of cephalopod nervous system complexity.

To support their findings Authors approached a compelling series of analyses, aimed at describing topological and regulatortory genome organization in one cephalopod species, namely *Euprymna scolopes*. Authors also attempted a comparison with other selected metazoan species, to further support their findings.

Response: Thank you.

I found the manuscript easy to read in the great majority of cases. Sometimes it is affected by a lack of proper punctuation in several sentences that make it quite difficult to read. There are also some typo, thus I suggest that the Authors will read with attention the whole text - including Supplementary Notes - and revise their English in order to further improve the overall quality of the manuscript.

Response: Thank you, we have revised the text.

I have few notes and comments for the Authors to consider in a revision.

1. The results of in situ hybridization experiments (FISH and ISH) lack of a control image to show. I suggest that the authors extend the data shown in Fig. 4e by adding some magnification and/or additional images in a more comprehensive figure to be included as Supplementary Material. As shown figures are not very informative.

Response: We thank the reviewer for these comments. We have substantially reworked the in situ images and now have included a new Figure 5 demonstrating the expression in sections of wholemount colorimetric staining, including both a negative control as well as a positive control (alpha tubulin). These panels demonstrate strong, specific staining in the nervous system, as well as in visceral organs, which are also summarized in detailed schematics.

1.1. Scale bar is added in Fig. 4e, but there is no indication in the figure legend.

Response: We have modified and describe the scale bar (new Figure 5)

1.2 Authors should indicate the stage (they refer to late E. scolopes embryos).

Response: Thank you, we have now indicated that these embryos are stage 27-29 in the figure legend.

1.3 A control section/image for each of the genes considered is suggested to be added in the additional supplementary figure requested.

Response: We have added both negative and positive control images and provided additional sectioning/staining images both of tubulin and the other genes examined. We have included these additional images to demonstrate the specificity of our in situ hybridizations and to highlight the dynamic expression of genes within this cluster - for example, we see that the tRNA ligase and ceramide 1 binding protein are more highly expressed in the central brain, the integrator complex subunit and amyloid binding protein are more broadly expressed in the central brain, as well as in the optic lobes, and the in the visceral organs.

1.4. Supplementary Figure Extended Data Fig. 8 (ISH): the identification of areas within the brain is not possible through the image; Authors need to identify better areas, even though this is an embryonic brain, considering that Authors are using late embryos (as reported) it is possible to identify specific areas/masses within the so-called "brain". This will provide further relevance to the finding, despite the limitation in the number of genes selected.

Response: As described in our response to the above request, we have added additional images, including sections focused on the central brain with detailed annotation, to demonstrate the expression in the different brain lobes, We have added additional images and included a navigation panel.

1.5 This reviewer finds difficult to understand the rationale for using the five genes

Response: Thank you we have tried to improve the rationale behind the choice/focus on this particular microsynteny:

“Given this insight, we further sought to investigate the potential function of novel cephalopod microsyntenies in expression modules 2 and 4 that showed the highest contribution to neuronal tissue expression domains. They can thus be considered a useful test set of functional microsyntenies that were involved in the evolution of the cephalopod nervous system. We thus examined the genomic rearrangement, the regulatory landscape and expression of genes within one of the representative cephalopod-specific microsyntenies from expression module 2. It was one of the clusters with the highest number of genes, encoding for ceramide-1 phosphate transfer protein, phenylalanine-tRNA ligase, splicing factor 3B subunit, integrator complex subunit and amyloid protein binding protein.”

1.6 in the Supplementary Figure (Extended Data Fig. 8) legend abbreviations refer to structures that are not depicted or where no signal is observed.

Please amend all Figures, Legends accurately

Response: We have reworked this section and included a new main Figure to illustrate the expression pattern. The fluorescent images are now in Extended Data Figure 10.

1.7 Overall the data shown after the included in situ hybridization experiments require a more accurate description and a more accurate representation, including representative additional sections, and more detailed magnifications to better show specificity of signal observed; this also requires control sections to be shown.

Response: As described above, we have added several new panels and detailed views of our in situ hybridizations, including controls, in a new figure to address this. See new Figure 5.

2. Use of Live Animals and sample collection.

2.1 Despite Authors obtained live animals/embryos from labs in the USA, it is highly recommended to refer to ARRIVE checklist (<https://arriveguidelines.org/arrive-guidelines>) that Nature Journals advice to follow.

Many details are missing in this version of the manuscript.

It is strongly advised to modify the relevant sections.

2.2 Experiments and live animals obtained not in EU are not following Directive 2010/63/EU.

Nevertheless, following ARRIVE checklist and in general considering that Cephalopods are included in the Directive 2010/63/EU a statement about animal welfare and ethical compliance of animals care should be provided.

Response: We thank the reviewer for pointing out this important shortcoming of our manuscript. In our work both in the EU and the US, we used exclusively embryonic stages that do not fall under the Directive 2010/63/EU. We nevertheless applied the latest anaesthesia protocols listed in the manuscripts we cited. Rearing of animals was conducted in Vienna Zoo and MBL . We have additionally specified in the text that all work was performed in compliance with the EU Directive 2010/63/EU on cephalopod use and AAALAC guidelines on the care and welfare of cephalopods.

3. Minor:

3.1 Supplementary Notes #1. Authors should refer to Collins, A.J., and Nyholm, S.V. (2010). Obtaining Hemocytes from the Hawaiian Bobtail Squid *Euprymna scolopes* and Observing their Adherence to Symbiotic and Non-Symbiotic Bacteria. *J Vis Exp* (36), e1714. when referring to "anaesthesia"

applied to these animals and to CSHL Protocols available for the species.

The reference to Fiorito et al., 2015 (Guidelines for animal care) is a general one; it maybe utilized here only to refer to the slow increase of solutes to reach the final concentration in the anaesthetic solution.

Response: We thank the reviewer for pointing out this discrepancy. The reviewer is correct that we cited Fiorito et al. 2015 to highlight general recommendations for handling cephalopods. We have therefore added references to Collins and Nyholm (2010), as well as Butler-Struben et al. (2018), which demonstrates the anesthetic effects of the protocol we used, and Shigeno et al. (2015), which better describes our approach.

3.2 "p" are indicated in several instances; a sentence or a reference to the statistics adopted to obtain that p value will be helpful for the Reader

(somewhere in the text, summarize statistics; it is diluted in the text somewhere and is not clear enough).

Only in one figure legend I can trace

reference to statistical tests adopted

Response: We have added mentions to the statistical tests used

3.3 Lines 137-141 Main text; the sentence reads not very clear.

Response: Thank you, we have revised the text.

3.4 I am unable to trace BioProject data, Authors declare to being deposited. Please amend the sentence or correct UID

Response: We currently have the data deposited as available to reviewers in GEO repository

GEO accession GSE157549 can be accessed via

<https://www.ncbi.nlm.nih.gov/geo/query/acc.cgi?acc=GSE157549>

with token sxscmskzlmjfi

We will make data publicly available upon paper acceptance.

3.5 Supplementary Notes

a. Please use headings of tables repeating in the following pages to facilitate readability

Response: We have attempted to improve table placement

b. Table 2: sort by taxonomy; add taxa (these does not necessarily match with the logic of inclusion of Table 3)

Response: We have sorted by taxonomical grouping as suggested

c. Table 3: includes/mention Rotifera in the Lophotrochozoa, please correct

Response: We have used the more appropriate term of "Spiralia"

d. better justification for not using hemocytes data is required. The low abundance of transcripts is novel/surprising to this reviewer, and requires justification

Response: We excluded hemocyte data because of very low total read count in that tissue, affecting estimates of co-expression and variation.

e. Table 6 indicate F/R

Response: Thank you, we have included forward/reverse information for primers in Table 6

f. beta-tubulin relative abundance in tissues considered should be mentioned somewhere in supplementary material at the late embryo stage considered.

It is not always homogeneous in tissues to be reported as "control"

Response: We now provide expression data for beta-tubulin in Supplementary Tables 7 and 8.

Minor

Reviewer #4 (Remarks to the Author):

As an evolutionary biologist, I reviewed this manuscript in the context of broad patterns of evolution and implications for cephalopod genomics.

Overall, I think this is a very well written paper with rigorous analyses to support the author's major hypotheses. I spent more time re-evaluating the paper to determine if I could provide any additional useful critique - I could not.

Response: Thank you

My only significant comment relates to the introduction:

Line 47-49: This paragraph is oddly worded. It would be helpful to the reader to have it re-phrased as it left me unclear about the current status of this type of knowledge in cephalopods. E.g.

- Has evidence for genome breakage and addition of novel microsyntenies in cephalopods been published? This statement currently reads as this information is pre-existing. I cannot recall if this work was addressed in the two cephalopod genome papers cited?

Response: Loss of microsyntenies was suggested in the octopus genome paper and gain of microsyntenies was only possible to estimate with the addition of the bobtail squid genome. However, those studies lacked the phylogenetic sampling and statistical assessment presented here.

- Same comment for the concluding sentence. Cite any previous work where this type of question was addressed. Have the extent and impact of arrangements not been investigated, or did data/etc. limitations make asking such questions impossible? This seems an important point to emphasize. It would better frame the significant contribution this work provides to the field of cephalopod genomics.

Response: Thank you, we have reworked our conclusion paragraph (and improved on introduction).

REVIEWER COMMENTS

Reviewer #1 (Remarks to the Author):

The revised version of the manuscript by Schmidbaur et al is a substantial improvement over the original submission. While the paper is still more condensed than it needs to be, it is much more readable than before, and some of my original concerns have been addressed adequately. However, there still remain several issues that are either unclear or not yet supported by sufficiently strong arguments:

- It is a bit confusing that, while the authors claim that the cephalopod synteny blocks are enriched for genes with neural function (thereby indicating cephalopod nervous system innovations), it is not visible from the GO analysis. On the contrary, while the metazoan synteny blocks are associated with overrepresentation with several terms normally associated with neurons and nervous system development (neurotransmitter transport, synaptic vesicle endocytosis, Wnt signalling), the terms that come up with cephalopod synteny blocks are practically without exception housekeeping. Is it because cephalopod genes do not have good GO annotation and neural term annotations are not consistently present in their mammalian orthologs, or is there something else that is the problem?

- If there are many neural genes in cephalopod microsynteny blocks, it is quite unusual that those blocks are so transcriptionally active - shouldn't they be silent outside the nervous system? Or is it because of "bystander" genes which are by and large ubiquitously expressed?

- One observation in favour of cephalopod microsynteny blocks as new GRB-like regions of long-range regulation is the fact that there are few intergenic spaces between them. The GRB model predicts that each round of whole genome duplication (WGD) will result in more developmental genes being retained in two copies than the adjacent housekeeping genes, which rediploidise more often. The result of several rounds of WGD is thinning out of housekeeping genes around the developmental genes, and their original locations remaining as gene deserts that harbour enhancers (and CNEs) that were ancestrally in introns of housekeeping genes. Indeed, many intronic enhancers in housekeeping genes that control a neighbouring developmental gene in mammals are in a gene desert in teleost genomes, where a housekeeping gene used to be (e.g. Kikuta et al 2007). In *Amphioxus*, which had two fewer WGDs than most vertebrates and has a more ancestral arrangement of GRBs, many CNEs are in introns of housekeeping genes while their orthologs are in gene deserts in vertebrates. So, if a gene that was originally not under long-range developmental regulation was to acquire long-range regulation and CNEs, it would most likely start as close to other genes, and its new long-range enhancers would colonise any sequence within the "striking range" of its promoters. Since intergenic distances were originally short, a high proportion of the new elements would be within introns of the target gene and adjacent ubiquitously expressed genes, locking them into syntenic arrangement and turning them into bystanders. The short intergenic sequences would then mean there were no WGDs in cephalopods since those new regulatory blocks were established. If true, that would be a very elegant model to explain the observations at these loci.

- "Surprisingly, we found that novel microsynteny blocks are more likely to form tight interaction regions, reflected by subtrees with few branches, when compared to ancestral synteny blocks or randomly sampled synteny blocks (Figure 3b)" - This is not what Figure 3b shows - it shows that the difference between cephalopod and metazoan synteny blocks is not significant.

- "Three-dimensional models revealed that novel cephalopod microsynteny blocks have distinct spatial properties from ancient microsynteny blocks." - Since cephalopod microsynteny blocks have more ubiquitously expressed genes and a smaller proportion of intergenic regions, could it be that it is the ubiquitously expressed genes that sort into the active "A" compartment result in the observed differences?

- The CNE analysis is not quite satisfactory. First, I had to dig deep through the supplementary material to find how the CNE sets were produced, and I still could not find key information 1) what is the minimum CNE size that was considered, and 2) what is the (pairwise) conservation level. These two parameters have to be tuned separately for each pairwise comparison to reveal similar CNE density patterns across varying distances. Also, the easiest way to see if anything like GRBs defined by CNEs are present is to visualise the CNE densities along the chromosome (like in Engstrom et al, 2008. Ancora: a web resource for exploring highly conserved noncoding elements and their association with developmental regulatory genes. Genome Biology 9: R34.). In this paper the authors seemed to use the same parameters (100bp and unknown level of similarity) both between squid and octopus and between two squids, resulting in a sparse set in the former case, and a large set but potentially with non-CNE alignments included. I would suggest to produce CNEs at several combinations of the two parameters, and to visualise their densities in browser along each other, to immediately spot regions of high density and inspect their gene content and arrangement, and try to answer the original questions again.

- It is not expected that all CNEs have the properties of active enhancers in every developmental stage, so their weak overlap with ATAC signal (a property shared with CNEs of vertebrates in most tissues and developmental stages) does not mean that their regulatory role is in question.

Minor:

- Line 27: "The genomic bases" - should be "The genomic basis".

- Line 81-82: " Our open chromatin data reveals regions potentially accessible to transcription factors and thus constituting regulatory elements." - This sounds misleading - the authors probably meant "Our open chromatin data reveals regions accessible to transcription factors and thus potentially constituting regulatory elements." Not all accessible chromatin are regulatory elements, and the current sentence makes it sound like they are.

- Line 317-318: "in the intron of phenylalanine-tRNA (Figure 4b)" should be "in the intron of phenylalanine-tRNA ligase (Figure 4b)".

Reviewer #2 (Remarks to the Author):

This new version of the manuscript by Schmidbauer et al is significantly improved and addresses all the points raised in my previous revision. The text reads much better and the authors have now included substantial new analyses that solidify key of their previous statements and conclusions, opening a new avenue to understand the genomic mechanisms underpinning the evolution of animal morphological innovations.

Best wishes,

Chema Martin

Reviewer #3 (Remarks to the Author):

The current version of the manuscript is improved a lot. Only an additional note, Fig 5 and Suppl Fig 10 - data presented with limited comments and not adequately referring/justifying the relative distribution of MACI genes considered when compared with tubulin (chosen by Authors as other 'ref gene')

Maybe the Authors want to simply provide a general expression of broad distribution of selected genes
(the reason why these are chosen is not clearly indicated)

REVIEWER COMMENTS

Reviewer #1 (Remarks to the Author):

The revised version of the manuscript by Schmidbaur et al is a substantial improvement over the original submission. While the paper is still more condensed than it needs to be, it is much more readable than before, and some of my original concerns have been addressed adequately. However, there still remain several issues that are either unclear or not yet supported by sufficiently strong arguments:

Response: Thank you!

- It is a bit confusing that, while the authors claim that the cephalopod synteny blocks are enriched for genes with neural function (thereby indicating cephalopod nervous system innovations), it is not visible from the GO analysis. On the contrary, while the metazoan synteny blocks are associated with overrepresentation with several terms normally associated with neurons and nervous system development (neurotransmitter transport, synaptic vesicle endocytosis, Wnt signalling), the terms that come up with cephalopod synteny blocks are practically without exception housekeeping. Is it because cephalopod genes do not have good GO annotation and neural term annotations are not consistently present in their mammalian orthologs, or is there something else that is the problem?

Response: Our proposal that cephalopod microsynteny blocks mainly contribute to neuronal expression is based on the RNA-seq and in-situ expression analysis of genes within microsynteny blocks (Figures 4 and 5). Since GO term prediction is based on homology with other species, it is not unexpected that some of the cephalopod microsynteny GO terms are broad, yet can be associated with neuronal function via expression (e.g., mRNA cleavage of one of the genes in Figure 5). The reviewer is also right that GO term enrichment can be biased by the bystander nature of cephalopod microsynteny blocks. Therefore, we see the GO term enrichment analysis and in situ/RNAseq profiling as two complementary methods reflecting the special nature of cephalopod microsynteny blocks.

- If there are many neural genes in cephalopod microsynteny blocks, it is quite unusual that those blocks are so transcriptionally active - shouldn't they be silent outside the nervous system? Or is it because of "bystander" genes which are by and large ubiquitously expressed?

Response: In our analyses cephalopod microsynteny blocks do mainly contribute to the expression modules 2 and 4 which have a strong "neuronal" component. Since the cephalopod nervous system is distributed throughout the body, we do expect neuronal genes to also show up in tissues such as arms (which contain two thirds of the nervous system). The bystander genes (as is in the case of the microsynteny block shown in Figure 5) we predict to be rather general enzymes expected to be broadly expressed.

- One observation in favour of cephalopod microsynteny blocks as new GRB-like regions of long-range regulation is the fact that there are few intergenic spaces between them. The GRB model predicts that each round of whole genome duplication (WGD) will result in more developmental genes being retained in two copies than the adjacent housekeeping genes, which rediploidise more often. The result of several rounds of WGD is thinning out of housekeeping genes around the developmental genes, and their original locations remaining as gene deserts that harbour enhancers (and CNEs) that were ancestrally in introns of housekeeping genes. Indeed, many intronic enhancers in housekeeping genes that control a neighbouring developmental gene in mammals are in a gene desert in teleost genomes, where a housekeeping gene used to be (e.g. Kikuta et al 2007). In *Amphioxus*, which had two fewer WGDs than most vertebrates and has a more ancestral arrangement of GRBs, many CNEs are in introns of housekeeping genes while their orthologs are in gene deserts in vertebrates. So, if a gene that was originally not under long-range developmental regulation was to acquire long-range regulation and CNEs, it would most likely start as close to other genes, and its new long-range enhancers would colonise any sequence within the "striking range" of its promoters. Since intergenic distances were originally short, a high proportion of the new elements would be within introns of the target gene and adjacent ubiquitously expressed genes, locking them into syntenic arrangement and turning them into bystanders. The short intergenic sequences would then mean there were no WGDs in cephalopods since those new regulatory blocks were established. If true, that would be a very elegant model to explain the observations at these loci.

Response: Thank you for this comment. This is indeed what is expected and confirmed by our observation - cephalopod genomes, despite their size, lack the two rounds of whole genome duplications and thus their microsynteny blocks that abide by the GRB model should show shorter intergenic distance (no gene deserts due to gene loss after whole genome duplication) and regulatory sequences (as predicted by ATAC signal) in intronic regions.

- "Surprisingly, we found that novel microsynteny are more likely to form tight interaction regions, reflected by subtrees with few branches, when compared to ancestral syntenies or randomly sampled syntenies (Figure 3b)" - This is not what Figure 3b shows - it shows that the difference between cephalopod and metazoan syntenies is not significant.

Response: Thank you, we have corrected the sentence

- "Three-dimensional models revealed that novel cephalopod microsynteny have distinct spatial properties from ancient microsynteny." -

Since cephalopod microsynteny has more ubiquitously expressed genes and a smaller proportion of intergenic regions, could it be that it is the ubiquitously expressed genes that sort into the active "A" compartment result in the observed differences?

Response: We have attempted to do this analysis, however the lack of methylation/histone modification data is really needed to properly assign A/B. So far, we could not find any evidence for synteny association to A/B. The following paragraph was added:

The GC content of metazoan and cephalopod-specific microsynteny was evaluated along with prediction of A/B compartments based on the Hi-C interaction matrix (Online Methods). The analysis did not provide sufficient evidence for one of the microsyntenic types being more prevalent in A (or B) compartment. Until further experimental data are available (such as methylation and acetylation profiling) for *Euprymna scolopes*, we assume both cephalopod-specific and metazoan microsynteny are similarly distributed within A/B compartment.

- The CNE analysis is not quite satisfactory. First, I had to dig deep through the supplementary material to find how the CNE sets were produced, and I still could not find key information 1) what is the minimum CNE size that was considered, and 2) what is the (pairwise) conservation level. These two parameters have to be tuned separately for each pairwise comparison to reveal similar CNE density patterns across varying distances. Also, the easiest way to see if anything like GRBs defined by CNEs are present is to visualise the CNE densities along the chromosome (like in Engstrom et al, 2008. Ancora: a web resource for exploring highly conserved noncoding elements and their association with developmental regulatory genes. Genome Biology 9: R34.). In this paper the authors seemed to use the same parameters (100bp and unknown level of similarity) both between squid and octopus and between two squids, resulting in a sparse set in the former case, and a large set but potentially with non-CNE alignments included. I would suggest to produce CNEs at several combinations of the two parameters, and to visualise their densities in browser along each other, to immediately spot regions of high density and inspect their gene content and arrangement, and try to answer the original questions again.

Response: We have now run additional analyses and added a new Supplementary Figure 9 and Supplementary Table 5 that shows overall consistency between different sizes and similarity thresholds. Finding putative CNEs is still problematic in cephalopod genomes, due to the lack of genomic sampling of sufficient quality. Overall our datasets provide for a useful first step in exploring this diversity. As the reviewer suggested, we now also plot putative CNE positions as density along the chromosome (Supplementary Figure 9) and furthermore provide a note in the manuscript about genome browser availability (<http://metazoa.csb.univie.ac.at:8000/euprymna/jbrowse> or upon request) where the putative CNEs can be further explored.

- It is not expected that all CNEs have the properties of active enhancers in every developmental stage, so their weak overlap with ATAC signal (a property shared with CNEs of vertebrates in most tissues and developmental stages) does not mean that their regulatory role is in question.

Response: We agree, we keep our phrasing as-is to simply state our observation.

Minor:

- Line 27: "The genomic bases" - should be "The genomic basis".

Response: corrected

- Line 81-82: " Our open chromatin data reveals regions potentially accessible to transcription factors and thus constituting regulatory elements." - This sounds misleading - the authors probably meant "Our open chromatin data reveals regions accessible to transcription factors and thus potentially constituting regulatory elements." Not all accessible chromatin are regulatory elements, and the current sentence makes it sound like they are.

Response: corrected

- Line 317-318: "in the intron of phenylalanine-tRNA (Figure 4b)" should be "in the intron of phenylalanine-tRNA ligase (Figure 4b)".

Response: corrected

Reviewer #3 (Remarks to the Author):

The current version of the manuscript is improved a lot. Only an additional note, Fig 5 and Suppl Fig 10 - data presented with limited comments and not adequately referring/justifying the relative distribution of MACI genes considered when compared with tubulin (chosen by Authors as other 'ref gene')
Maybe the Authors want to simply provide a general expression of broad distribution of selected genes (the reason why these are chosen in not clearly indicated)

Response: We have selected the genes for Figure 5 in situ expression analysis because they constitute the largest cephalopod-specific microsyntenic cluster. We also provide analysis and annotation of the general expression pattern of all cephalopod and metazoan syntenies (Figure 3) as well as new supplementary data 1, listing their annotations and genomic locations. We have also clarified the use of beta-tubulin in the figure legend.

REVIEWERS' COMMENTS

Reviewer #1 (Remarks to the Author):

The authors have responded to some of the issues I raised in my second review, and showed some resistance to others. The latter is a pity, because I believe it would have resulted in a better and more interesting paper. However, in order not to be “that” reviewer any more than I already am, I will just let it go, because apparently both the editor and the other two referees think the paper is now ready to publication. Only two minor things:

- I don't agree with the new sentence “Until further experimental data are available (such as methylation and acetylation profiling) for *Euprymna scolopes*, we assume both cephalopod-specific and metazoan microsynteny are similarly distributed within A/B compartment.”. Most genomic features are NOT similarly distributed between A and B compartments, and there is little reason to adopt it as prior belief. It would be better if the authors just stated that they just don't know, based on the data available at the moment.

It is great that the authors have provided a genome browser with CNE tracks at varying combinations of minimum length and sequence conservation. I was a bit disappointed with gene annotation in the browser (or lack of) though. For example, in this screenshot

The left sidebar contains the 'Available Tracks' section, which is used to filter and select the tracks shown in the main window. The 'Gene Annotation PASA' track is selected, and the gene 'asmb1_19087.p1' is highlighted in the main track.

when you click on a gene in Gene Annotation PASA track (here asmb1_19087.p1), you get

gene asmbL_19087.p1

Primary Data

- Name** asmbL_19087.p1
- Type** gene
- Position** Lachesis_group39__38_contigs__length_50967332:25495723..25532069 (- strand)
- Length** 36,347 bp

Attributes

- id** gene17257
- seq_id** Lachesis_group39__38_contigs__length_50967332
- source** transdecoder
- uniqueID** offset-557172981

Region sequence

```
>Lachesis_group39__38_contigs__length_50967332
Lachesis_group39__38_contigs__length_50967332:25495723..25532069 (- strand) class=gene length=36347
ATGGTCAAAGCATAACAACGGTCAAAGCATAcgactgtcaaagcatacgcaggtcaaagca
tacaacggtcaaagcatacaacggtcaaagcatacgcaggtcaaagcatacgcactgtcaa
agcatacgcaggtcaaagcatacgcaggtcaaagcatacaacggtcaaagcatacaatgg
TCAAAGCATACGACGGTCAAAGCATACGATGGTCAAAGCATAACAACGGTCAAAGCATATA
ACGGTCAAAGCATACGACGGTCAAAGCATACGACGGCCGAAGCAAACGACGGTCAAACCA
TACGACTGTCAAAGCATACGACTGTCAAAGTATACAACGGTCAAAGCATACGACGGTTCA
TGCATACGACGGTCAAAGCATACAACGGCCAAAGCATACGACGGTCAAAGCATACGACTT
TCAAAGCATACTATGGTCAAAGCATACAACGGTCAAAGCATACGACTGTCAAAGCATACAA
```

Subfeatures

Primary Data

- Name** mRNA17257
- Type** mRNA
- Position** Lachesis_group39__38_contigs__length_50967332:25495723..25532069 (- strand)

which only lists the sequence and positions of exons of the gene, and provides no functional annotation. This annotation is apparently available in the database, because when you search using a gene name, you can retrieve them together with asmbL gene ID and description; here are the results of search for “SOX2”:

EGGNOG ANNOTATION

DOWNLOAD(.CSV)

gene ID	eggno seed ortholog	value	description	preferred name	pfams
asmbL_12065.p1	51511.ENSCSAVP00000010906	5.5e-11	thiol oxidase activity	QSOX2	Evr1_Alr,Thiorec
asmbL_12066.p1	51511.ENSCSAVP00000010906	1.9e-10	thiol oxidase activity	QSOX2	Evr1_Alr,Thiorec
asmbL_1381.p1	126957.SMAR011344-PA	1.9e-65	Transcription factor	SOX2	HMG_box,SOXp
asmbL_1382.p1	126957.SMAR011344-PA	2.3e-65	Transcription factor	SOX2	HMG_box,SOXp
asmbL_2445.p1	6500.XP_005108229.1	2.8e-61	DNA binding	SOX21	HMG_box,SOXp

This is the information that should be presented when clicking on a gene in the browser - it is so much more useful for most people than immediate access to exon sequences.

I tried to search manually using the gene ID "asmbL_19087.p1" here

Not Secure — metazoa.csb.univie.ac.at

Euprymna Genome Browser | Lachesis_group24_36_cont... | Lachesis_group39_38_conti... | Lachesis_group39_38_conti...

Euprymna Genome Portal

Main Jbrowse Blast Query Neo4J Datasets Contact

Query gene annotation and atac data by either gene ID or genomic position.

GENE ID BY ANNOTATION **ANNOTATION AND ATAC DATA BY GENE ID** ATAC DATA BY COORDINATE

CNES DATA BY COORDINATE

Gene ID
asmb1_19087.p1

padding (bp)
5000

Atac seq dataset
All

SUBMIT

but the search returned an error, with this or other IDs I tried:

As a result I still don't know the functional annotation of `asmb1_19087.p1`, or if it exists. Of course, I could stitch the exon sequences and blast them, but that would be asking users to do too much.

Response to reviewer 1: Thank you very much for your feedback.

– I don't agree with the new sentence "Until further experimental data are available (such as methylation and acetylation profiling) for *Euprymna scolopes*, we assume both cephalopod-specific and metazoan microsynteny are similarly distributed within A/B compartment.". Most genomic features are NOT similarly distributed between A and B compartments, and there is little reason to adopt it as prior belief. It would be better if the authors just stated that they just don't know, based on the data available at the moment.

Response: We have modified the sentence accordingly

– It is great that the authors have provided a genome browser with CNE tracks at varying combinations of minimum length and sequence conservation. I was a bit disappointed with gene annotation in the browser (or lack of) though. For example, in this screenshot

Response: Thank you. While the browser and the resource page functionality was not intended to be part of this manuscript, in the future we will introduce more user-friendly query system for annotations and other genomic information.